# Language Model Alignment with Elastic Reset

**Michael Noukhovitch**[*]
Mila, Université de Montréal

**Samuel Lavoie**
Mila, Université de Montréal

**Florian Strub**
Google Deepmind

**Aaron Courville**
Mila, Université de Montréal
CIFAR AI Chair

## Abstract

Finetuning language models with reinforcement learning (RL), e.g. from human feedback (HF), is a prominent method for alignment. But optimizing against a reward model can improve on reward while degrading performance in other areas, a phenomenon known as reward hacking, alignment tax, or language drift. First, we argue that commonly-used test metrics are insufficient and instead measure how different algorithms tradeoff between reward and drift. The standard method modified the reward with a Kullback-Lieber (KL) penalty between the online and initial model. We propose **Elastic Reset**, a new algorithm that achieves higher reward with less drift without explicitly modifying the training objective. We periodically reset the online model to an exponentially moving average (EMA) of itself, then reset the EMA model to the initial model. Through the use of an EMA, our model recovers quickly after resets and achieves higher reward with less drift in the same number of steps. We demonstrate that fine-tuning language models with Elastic Reset leads to state-of-the-art performance on a small scale pivot-translation benchmark, outperforms all baselines in a medium-scale RLHF-like IMDB mock sentiment task and leads to a more performant and more aligned technical QA chatbot with LLaMA-7B. Code available at github.com/mnoukhov/elastic-reset.

## 1 Introduction

Dialogue agents that can effectively interpret and use language are a long-term challenge for NLP. The rise of large pretrained language models (LMs) [Brown et al., 2020] made language model finetuning one of the most promising research directions to achieving capable dialogue agents [Bender and Koller, 2020]. Recently, reinforcement learning (RL) has become a key ingredient of finetuning large LMs for interaction with humans [Ziegler et al., 2019, Ouyang et al., 2022, Bai et al., 2022], notably shown in ChatGPT [OpenAI, 2022]. A reward model is learned on the alignment objective, such as learned human preferences [RLHF; Christiano et al., 2017, Stiennon et al., 2020], and the language model is finetuned to optimize the reward. But training on the RL objective moves the model away from its pretraining and can reduce performance on important benchmarks [Ouyang et al., 2022] and even drifting away from natural language syntax and semantics [Lazaridou et al., 2020].

"Language drift" [Lee et al., 2019, Lazaridou et al., 2020], "alignment tax" [Askell et al., 2021], "reward model overoptimization" [Gao et al., 2022], or LM-specific "reward-hacking" [Clark and Amodei, 2016] is inherent to RLHF. In the extreme case, models learn to achieve high reward by generating nonsense text that is unintelligible to humans [Lewis et al., 2017]. Methods to mitigate this issue range from re-running pretraining [Lowe* et al., 2021], grounding in other modalities [Lee et al.,

---

[*]Correspondance to `michael.noukhovitch@umontreal.ca`

37th Conference on Neural Information Processing Systems (NeurIPS 2023).

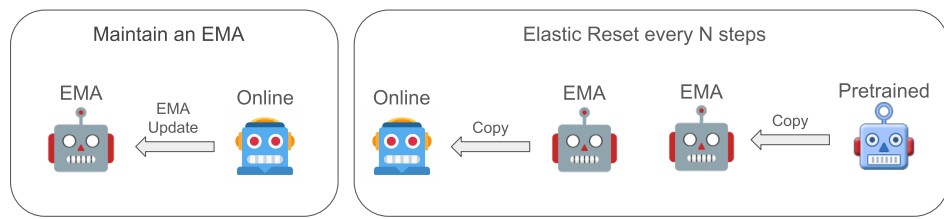

Figure 1: Elastic Reset. In actor-critic RL, we reset the policy but maintain the value function.

2019], masking the LM generation [Ramamurthy et al., 2022] and iterated learning [Lu et al., 2020]. But the standard, and by far most popular approach, adds a Kullback-Lieber (KL) divergence penalty to the reward in order to prevent the finetuned model from drifting too far from the pretrained model [Jaques et al., 2017, 2019, Ziegler et al., 2019]. Still, all methods are insufficient over a large-enough training horizon so models are early-stopped before reaching a catastrophic level of drift.

Gao et al. [2022] find that achieving reward is proportional to drift from the initial model, but that not all drifts are equal. We wish to make small but effective changes that achieve high reward but maintain capabilities, yet auxiliary losses such as the KL penalty don't seem improve this tradeoff and only serve to slow down training [Gao et al., 2022]. We posit that RLHF training requires a useful inductive bias that does not modify the training objective. Inspired by recent work in generalization for image classification [Zhou et al., 2022], sample-efficient RL [Nikishin et al., 2022, D'Oro et al., 2023], and inducing compositional language [Li and Bowling, 2019], we propose to use resets. Similar to iterated learning [Kirby, 2001], iteratively resetting a model has been shown to reduce overfitting in language and RL scenarios [Rita et al., 2022]. In this work, we show iteratively resetting a model also reduces drift while attaining equal or better reward than just a KL penalty.

Unlike previous work in sample-efficient RL [Nikishin et al., 2022], RLHF is typically on-policy so it does not maintain a replay buffer with which to bootstrap learning after a reset. In lieu of a replay buffer, we reset the policy but maintain the value function. Yet resetting the policy to its initial state can still cause a large drop in performance. So we propose resetting to a model in-between our online and initial state, specifically to an exponential moving average (EMA) of our online policy, as EMA has been shown to be highly performant [Caron et al., 2021]. We still expect our EMA model to slowly drift, so we add a second step where we reset the EMA model to the initial model. We call this overall method **Elastic Reset** and illustrate it in Figure 1. Elastic Reset is implemented on top of regular RL methods such as REINFORCE [Williams, 1992] or PPO [Schulman et al., 2017].

First, we test our method on a small scale task: pivot translation with a transformer. In this classic benchmark for drift, we outperform all previous baselines and demonstrate state-of-the-art performance. Next, we re-evaluate how performance is measured in the field and argue for a metric of how each method trades off performance vs drift. We propose the Pareto Frontier Graph, a graphical measure that illuminates the trade-off between performance and drift and demonstrate that Elastic Reset dominates the baselines against this trade-off. Then, we scale up slightly to GPT2 and work on a popular task closer to RLHF, IMDB mock sentiment. Comparing to all baseline methods, we again show state-of-the-art performance on the benchmark. Through ablations, we show that Elastic Reset is robust to choices of hyperparameters, even more so than baselines. Finally, we scale up even more to true RLHF finetuning of Llama-7B in order to create a helpful technical QA chatbot using a StackExchange dataset. We again outperform the baseline, demonstrating how Elastic Reset mitigates the alignment tax while better optimizing the human feedback reward.

## 2    Related Work

"It is often difficult or infeasible to capture exactly what we want an agent to do, and as a result we frequently end up using imperfect but easily measured proxies" [Clark and Amodei, 2016]. In RL, this proxy is how we construct our reward and the consequence can be "reward-hacking" [Clark and Amodei, 2016]; an agent optimizes the reward but does not accomplishing the meaningful task. RLHF aims to align an agent with human preferences while maintaining the capabilities of the pretrained model, but uses a learned reward model as a proxy of human preferences [Christiano et al., 2017, Ziegler et al., 2019]. This can lead to LMs that optimize a reward model but degrade in

performance on general NLP benchmarks [Askell et al., 2021], overfit the reward model and do not generalize to true human preferences [Bai et al., 2022, Gao et al., 2022], or latch onto confounding factors hidden in the reward [Stiennon et al., 2020]. These effects are exacerbated if the reward model is updated during training such as in iterated RLHF [Bai et al., 2022] or related setups such as emergent communication [Lazaridou and Baroni, 2020], end-to-end dialogue [Lewis et al., 2017], learning RL policies through latent language [Andreas et al., 2018], and pivot translation [Utiyama and Isahara, 2007]. There, the phenomenon is known as "language drift" [Lee et al., 2019] and can lead to incoherent and unnatural linguistic outputs Lewis et al. [2017].

This phenomenon is inherent to RLHF. Gao et al. [2022] show that improvement on alignment / reward is proportional to drift from the initial model, but also find that different methods and design choices achieve different proportions of performance to drift. Therefore, a major challenge of RLHF is how to learn the reward in such a way as to minimize the drift, alignment tax, and reward-hacking. The standard approach used in most RLHF is to incorporate a KL penalty between the training language model and some fixed model [Jaques et al., 2019, Ziegler et al., 2019, Stiennon et al., 2020, Ouyang et al., 2022, Steinert-Threlkeld et al., 2022, Bai et al., 2022], usually the initial, pretrained model. Less common is to add the original pretraining task to the finetuning objective (termed S2P by Lowe* et al. [2021]) but this can be compute-intensive and requires maintaining the pretraining data which may be even more expensive for larger models [Brown et al., 2020]. On small-scale pivot-translation, Lu et al. [2020] propose iterated learning with student-teacher distillation but it too is relatively compute intensive. Recently, Ramamurthy et al. [2022] propose to maintain a delayed masking model and mask the LM to output only the top-$p$ tokens. Elastic Reset takes inspiration from both of these, using an iterated process and maintaining an EMA model. Apart from better performance, our method is more space efficient and maintains the EMA on CPU whereas both other methods require maintaining an extra model on GPU. It is also more compute efficient as resetting weights and EMA updates are very cheap operations, whereas Lu et al. [2020] requires a long distillation phase and Ramamurthy et al. [2022] requires an extra forward pass with the masking model. There exist less-popular methods that have been applied to similar issues in RL: prompt-tuning [Singh et al., 2022], using a fixed model to generate many options and re-ranking using a reward model [Lazaridou et al., 2020, Meta FAIR Diplomacy Team et al., 2022], and grounding the output in a separate modality [Lee et al., 2019] but none have been used for RLHF or at scale.

Our method is inspired by recent works that leverage resets for single agent RL [Nikishin et al., 2022, D'Oro et al., 2023], image classification [Zhou et al., 2022], and emergent communication [Rita et al., 2022]. Those works generally train from scratch and reset to random initializations in order to improve generalization. Our scenario requires resetting to pretrained models and focuses on improving the tradeoff between performance and drift from this pretrained model. Elastic Reset can be seen as an on-policy alternative to Nikishin et al.'s [2022] off-policy resets; whereas they maintain the old replay buffer, we maintain the value model and an EMA of our policy.

Finally, the pretrain-then-RL-finetune setup with the goal of maintaining pretrained knowledge can be seen as a two-step, RL-specific instance of continual learning and therefore language drift has links to catastrophic forgetting [McCloskey and Cohen, 1989]. There is a clear similarity between mitigation methods: rehearsal [Robins, 1995] or experience replay [Rolnick et al., 2019] is equivalent to multitasking with the pretraining objective [Lowe* et al., 2021] and weight-update regularization [Kirkpatrick et al., 2017] has similarities to KL regularization [Jaques et al., 2019].

## 3 Elastic Reset

The standard method against drift is a KL penalty, generally between the learning policy $\theta$ and the initial, pretrained model $\theta_0$. It is calculated empirically over the minibatch of training inputs $x$ and outputs $y$ and used as an auxiliary reward with coefficient $\beta$ on top of the regular reward model $r$

$$R(x, y) = r(x, y) - \beta \log \frac{\pi_\theta(y|x)}{\pi_{\theta_0}(y|x)} \tag{1}$$

For Elastic Reset, we maintain an exponential moving average $\bar{\theta}$ of our learning model $\theta$ and choose a decay hyperparameter parameter $\eta$. We initialize $\bar{\theta} \leftarrow \theta_0$ and after every online model step, we update our EMA model $\bar{\theta} \leftarrow (1 - \eta)\theta + \eta\bar{\theta}$. Every $n$ steps, Elastic Reset sets the online model to the

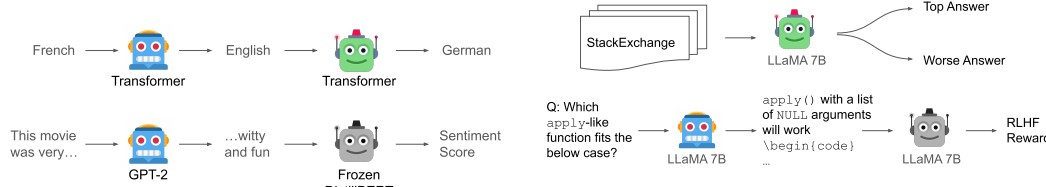

Figure 2: The Translation Game (top left), IMDB mock sentiment task (bottom left), and StackLLaMA (right). We show all RL finetuning setups and StackLLaMA's reward modelling (top right).

EMA model $\theta \leftarrow \bar{\theta}$ and sets the EMA model to the initial model $\bar{\theta} \leftarrow \theta_0$. As with other methods, Elastic Reset can be easily combined with a KL penalty.

## 4  Translation Game: Careful Comparison to SOTA

**Setup**  We first investigate the pivot-translation benchmark of Lee et al. [2019], which was previously popular for small-scale methods countering drift. Two translation models, French to English (FR→EN) and English to German (EN→DE), are pretrained on IWSLT [Cettolo et al., 2012]. Then, the models are finetuned on translating French to German through English (FR→EN→DE) but given only paired French and German data from Multi30k [Elliott et al., 2016, 2017] as shown in Figure 2. The models are not given English at finetune-time so the challenge is optimizing FR→DE while maintaining fluency in the intermediate English. Whereas larger benchmarks have only proxies for drift, we can exactly measure the performance degradation in our setup with the standard translation metric BLEU on a held-out FR→EN validation set. Similarly, we measure success on the task with the FR→EN→DE BLEU score. Each model is an encoder-decoder Transformer [Vaswani et al., 2017] with 6 layers and all experimental details are available in Appendix A.

**Baselines**  The EN→DE reward model is simply trained using cross-entropy between predicted and true DE. Our lower-bound baseline is FROZEN ENGLISH, we freeze the FR→EN model and only update the EN→DE model. This models is guaranteed not to drift, but also cannot reach the best possible performance. For that, we need to update FR→EN by backpropagating through the discrete EN tokens. We follow Lee et al. [2019] and train FR→EN using REINFORCE [Williams, 1992] to estimate the gradient. As our base model, we combine REINFORCE with an exponentially moving baseline and, as with previous work, add a loss for entropy regularization.

When both FR→EN and EN→DE are being updated, we tend to see reasonably large drift and we compare to the best previous methods that counter it on this benchmark. We follow Lee et al. [2019] to simulate the standard KL penalty method, KL PENALTY by training an LSTM LM on IWSLT English text and adding a KL penalty with $\beta = 0.05$ to regularize the FR→EN model. MULTITASK learning, re-training, or S2P [Lowe* et al., 2021], adds the supervised FR→EN objective on IWSLT pretraining data as an auxiliary task for the FR→EN model. Finally, we implement Seeded Iterated Learning [SIL; Lu et al., 2020], which alternates between $n$ finetuning steps and $m$ steps of teacher-student distillation. FR→EN and EN→DE "teacher" models are finetuned on the translation game, then each distills knowledge into "student" model of itself, and finally the students are initialized as teachers for the next iteration. ELASTIC RESET is implemented on top of REINFORCE with a very minimal KL penalty $\beta = 0.001$ and uses an EMA decay $\eta = 0.99$. We run all models for 50k updates and reset every 23k steps to get 2 resets / 3 iterations within a run. Hyperparameters may differ between methods, e.g. $\beta$, because we used a minimal search to find the best hyperparameters for each method.

**Experiments**  For each method, we run 5 seeds and plot the validation scores over training for the end-to-end task score, FR→EN→DE BLEU, and drift score, FR→EN BLEU, in Figures 3b, 3a respectively. Following Lee et al. [2019], we also show the final validation score in Table 1. As a sanity check, FROZEN ENGLISH does not drift but also does not achieve a very high task performance. In line with previous results [Lu et al., 2020], all learning models initially improve FR→EN performance, likely because models are quickly, semi-supervised adapting from their pretraining (IWSLT) to the distribution of the finetune dataset (Multi30k). Afterwards, they start to overfit on their objective and FR→EN performance degrades. REINFORCE achieves the best

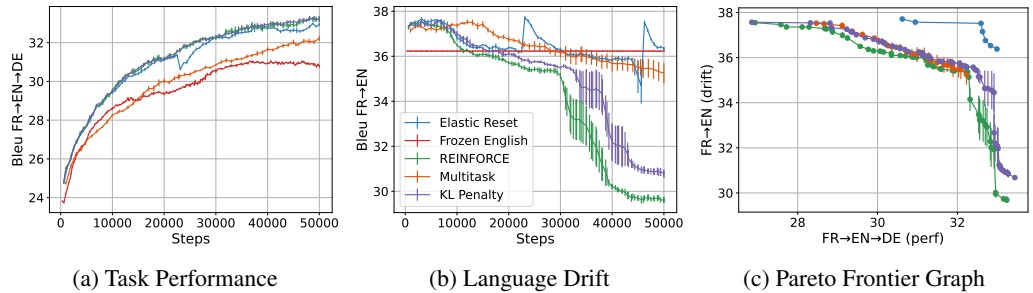

| (a) Task Performance | (b) Language Drift | (c) Pareto Frontier Graph |

Figure 3: Comparing Elastic Reset to all baseline methods on the Translation Game. We measure (a) Task Performance with FR→EN→DE BLEU and (b) Language Drift with FR→EN BLEU, on the validation set during finetuning. We plot the mean and standard error over 5 seeds. To compare how methods trade off the two metrics, we plot (c) the best achieved drift vs task performance.

Table 1: Translation Game final validation scores

|  | ↑ FR→EN→DE | ↑ FR→EN |
|---|---|---|
| FROZEN ENGLISH | 30.8±0.2 | **36.3±0.1** |
| REINFORCE | **33.2±0.3** | 29.6±0.3 |
| + SIL | 28.2±0.4 | 27.3±4.4 |
| + MULTITASK (S2P) | 32.2±0.3 | 35.2±1.0 |
| + KL PENALTY | **33.2±0.2** | 30.8±0.4 |
| + **ELASTIC RESET** | 32.9±0.1 | **36.3±0.1** |

possible task performance but drifts significantly. Despite extensive hyperparameter tuning and correspondence with the original authors, SIL does not manage to outperform the REINFORCE baseline so we exclude it from the figures for visual clarity but show values in Table 1 as well as full results in Appendix A. In line with previous work [Lee et al., 2019, Lu et al., 2020], we find that MULTITASK and KL PENALTY are both beneficial to reducing drift, but both represent a tradeoff. Whereas MULTITASK strongly reduces drift, it does not achieve a high task score. In contrast, KL PENALTY achieves a high task score but drifts quite drastically. Elastic Reset achieves nearly the best possible task score while maintaining the same drift score as the initial model. Visually, we see that our method track the baselines until the reset at 23k steps. After the reset, we see a slight performance drop but also a big jump back in terms of FR→EN drift. While the task performance recovers within 5k steps, the drift performance does not degrade to previous levels. For the second reset, the EMA model is slightly more drifted and so the reset is less pronounced for both task and drift, leading to faster task recovery but slightly more drift. Overall, Elastic Reset shows state-of-the-art results on the benchmark and outperforms all previous small-scale methods.

# 5   Pareto Frontier Graph

Simply evaluating validation curves side-by-side or looking at a table of final scores, it can be unclear which method is better if one drifts less but the other achieves a higher reward e.g. MULTITASK vs KL PENALTY in Table 1. Previous work on this [Lee et al., 2019] and other benchmarks [Ramamurthy et al., 2022] compare methods using simple point-estimates after training for a specific number of epochs. But this number of epochs is quite arbitrary as models never fully converge to a reward, they are early-stopped such that drift is not catastrophic. Since different setups may admit different levels of drift, we believe that evaluation should reflect the continuous tradeoff between task and drift. We extend Ziegler et al. [2019], and create a pareto frontier graph to plot each method's achieved task score vs drift metric on the validation set over training. We believe practioners will wish to choose the best model for some given task performance so, contrary to Ziegler et al. [2019], we plot the task score on x-axis and drift score on the y-axis. Improvement on a drift metric can either mean lower scores (perplexity) or higher scores (BLEU) so we always plot task score as increasing from bottom to top such that, graphically, a better method will functionally dominate a worse method. We plot the best achieved reward vs drift over all validation steps for the Translation Game in Figure 3c. Not only

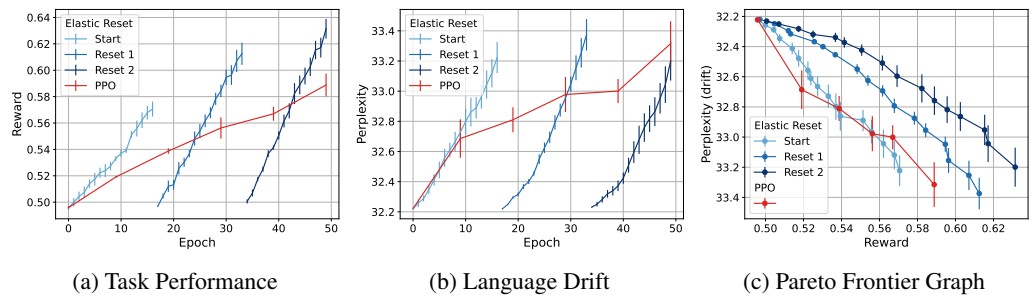

| | (a) Task Performance | (b) Language Drift | (c) Pareto Frontier Graph |

Figure 4: Plotting PPO vs Elastic Reset on IMDB but splitting the results visually between resets. We measure (a) Language Drift and (b) Task Performance via Semantic Score on the validation set over finetuning. All methods also include a KL penalty. We plot mean and standard error across 5 seeds.

Table 2: IMDB mock sentiment final test scores

| | ↑ SENTIMENT | ↓ PERPLEXITY |
|---|---|---|
| ZERO-SHOT | .489±0.01 | 32.45±0.13 |
| PPO | .596±0.02 | 33.45±0.40 |
| NLPO | .558±0.06 | 33.12±0.74 |
| **ELASTIC RESET** | .611±0.02 | 33.32±0.23 |

does Elastic Reset outperform the baselines at the final validation score, but it functionally dominates such that it is the best method for all levels of task performance it achieves.

## 6 IMDB Mock Sentiment: Ablation Study for RLHF

**Setup** Next, we scale to a larger benchmark that more closely approximates the standard RLHF setup. We use the recently released GRUE benchmark for RL training of LMs [Ramamurthy et al., 2022] and use IMDB mock sentiment [Ziegler et al., 2019], the main task where language models are susceptible to reward-hacking, shown in Figure 2. The goal is to complete an IMDB movie review with as positive a sentiment as possible. The baseline LM is GPT-2 [Radford et al., 2019] with 117M parameters further pretrained on the IMDB domain [Maas et al., 2011]. We learn a DistilBERT [Sanh et al., 2020] reward model on IMDB to output a sentiment score between 0 (negative) and 1 (positive). We then train our GPT-2 LM to complete different IMDB reviews while maximizing the sentiment reward. Following Ramamurthy et al. [2022], we measure reward-hacking / drift with our model's perplexity on the true IMDB data. If we consider knowledge of the IMDB data as a useful capability, then our initial model was finetuned on IMDB to maximize log-probability, i.e. minimize perplexity, and has the maximum capabilites. We measure divergence from the initial model, and decrease in capabilities, by the increase in our trained model's perplexity on ground truth IMDB data. In contrast to the previous task, a lower perplexity score corresponds to less drift.

**Baselines** Our main baseline is PPO [Schulman et al., 2017] with Ziegler et al. [2019] modifications for RLHF training, specifically adding a KL penalty with the frozen initial model (equivalent to KL WITH PRETRAINED) and dynamically decaying the coefficient $\beta$ over training. To further increase stability, Generalized Advantage Estimation [Schulman et al., 2015] is used for the advantage estimator. We also compare to NLPO [Ramamurthy et al., 2022], a recent method that extends PPO with a masking model to counteract drift. The masking model is initialized to the pretrained model and recieves delayed updates; it is set to the online model every $n$ steps. During training, the online model's output probabilities are restricted to the mask model's top $p$ tokens. We use the RL4LMs library Ramamurthy et al. [2022] and their default hyperparameters for both PPO and NLPO e.g. $\beta = 0.1$. We implement ELASTIC RESET on top of PPO with an EMA decay rate of 0.995 and greatly reduce the KL coefficient $\beta = 0.001$ to allow the model to drift more, then reset every 17 epochs such that we get two resets / three iterations during our training.

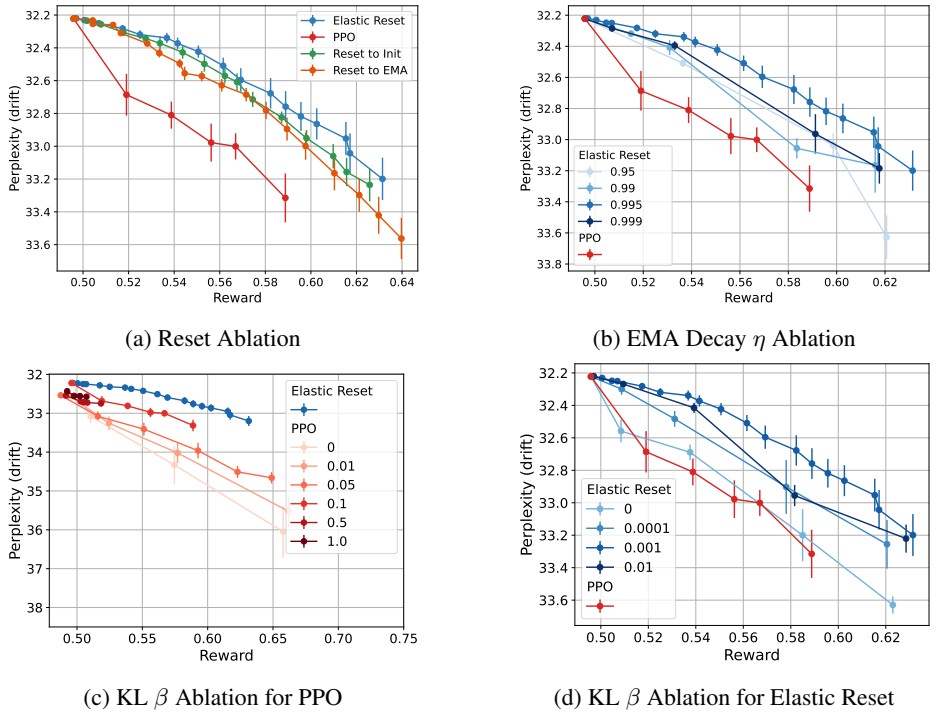

(a) Reset Ablation

(b) EMA Decay $\eta$ Ablation

(c) KL $\beta$ Ablation for PPO

(d) KL $\beta$ Ablation for Elastic Reset

Figure 5: Ablating Elastic Reset on the IMDB mock sentiment task. We plot pareto graphs using mean and standard error across 5 seeds.

**Experiments** We run all experiments for 5 seeds and report mean and standard error on our validation set for our reward, DistilBERT sentiment, and our drift score, perplexity. Following Ramamurthy et al. [2022], we run for 50 epochs (equivalent to 64k updates) and show our results in Figure 4. To make our resets more visible, we plot validation scores every epoch for Elastic Reset. Since the benchmark provides a test set as well, we compare all final models in Table 2. The PPO baseline performs quite well because it already includes a KL with the pretrained model. We find NLPO performs similarly to PPO, so we relegate NLPO results to Appendix B. Results from the original NLPO paper were stronger [Ramamurthy et al., 2022] but our reproduced numbers and curves were confirmed by the original authors [Ammanabrolu, 2023]. Elastic Reset achieves better semantic scores much faster by using a smaller KL penalty coefficient (0.001 vs PPO 0.1) but also drifts more to achieve them. As with the previous task, this drift is then mitigated by the reset and we see semantic task score improve in relation to drift over the iterations. Looking at the pareto graph in Figure 8c, we see that Elastic Reset far outpaces the baselines and provides a better tradeoff of reward vs drift for every reward.

**Ablations** We empirically investigate our method through ablations. Throughout this section we run experiments on IMDB with the same hyperparameters, unless otherwise mentioned. For brevity, we plot only the pareto graphs but include all other graphs in Appendix D.3 along with these same ablation experiments for the Translation Game, with similar results.

To investigate the source of improvement in Elastic Reset, we ablate the two resets: online to EMA, and EMA to initial model. We discard the second reset to get `Reset to EMA`: our model is reset to an EMA but the EMA is never reset. We also compare to the simplest reset idea, `Reset to Init`, and reset our policy to the initial model. We run all methods as previously and plot the pareto graph in Figure 5a, for task and drift graphs see Appendix D.3. We find that even simple resets are already performant but the two ablations have a tradeoff: `Reset to EMA` is better at lower reward because it maintains performance whereas `Reset to Init` does better at higher reward because it doesn't drift as much. Elastic Reset combines the benefits of both and outperforms each method.

Next, we consider our method's robustness to hyperparameters. First, we search along different EMA decay rates $\eta$ and plot our results in Figure 5b finding that our method is quite robust to choice of

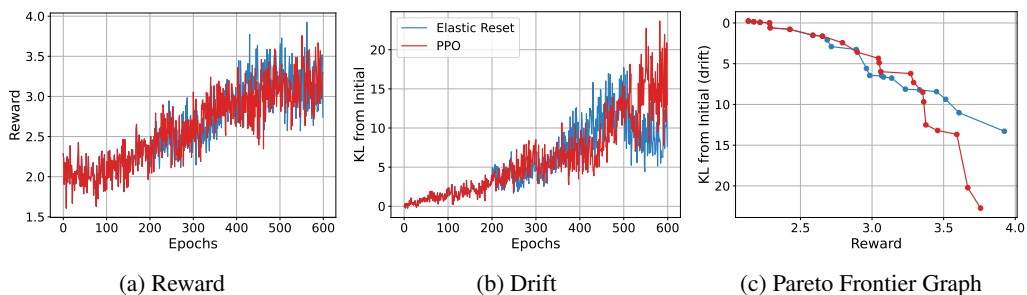

|                  |                  |                         |
| :--------------: | :--------------: | :---------------------: |
| (a) Reward       | (b) Drift        | (c) Pareto Frontier Graph |

Figure 6: Elastic Reset compared to PPO on StackLLaMA: A LLaMA-7B model RLHF finetuned on StackExchange as a helpful, technical QA chatbot

decay. Next, we investigate robustness to the choice of KL penalty coefficient $\beta$. We search across coefficients that range from 10x smaller to 10x larger than our best KL penalty coefficient for PPO ($\beta = 0.1$) and Elastic Reset ($\beta = 0.001$). For visual clarity, we plot PPO in Figure 5c and Elastic Reset in Figure 5d and only plot four points for PPO $\beta = 0, 0.01$ to maintain visual scale. We find that PPO is not robust to choice of KL and larger values correspond to better pareto curves but slower training. Results with NLPO are similar and shown in Appendix D.3. In contrast, Elastic Reset seems to be more robust to choice of KL with 0.001 producing the best curves while 10x larger and smaller values are similar. As opposed to PPO, Elastic Reset even works reasonably well without a KL penalty at all, ($\beta = 0$), matching PPO's best performance with a KL. This demonstrates that the expensive KL penalty may be replaced with the cheap Elastic Reset, although the combination of the two is best. This is also in line with previous work that have argued that the KL penalty may be unnecessary for RLHF [Bai et al., 2022, Gao et al., 2022]. We also ablate the frequency of resets in Appendix D.4.1 and find that pareto curves are essentially unchanged.

Finally, we provide an empirical intuition for Elastic Reset: in Appendix E.1 we show that resets iteratively improve the value function and in Appendix E.2 we show how EMA smoothes optimization but requires resetting in order to achieve high performance.

## 7  StackLLaMA: Practical RLHF

**Setup**   Finally, we apply Elastic Reset to a larger-scale RLHF pipeline. We choose LLaMA [Touvron et al., 2023] as it is a prominent open-source model that has demonstrated strong performance on benchmarks. We follow Beeching et al. [2023] to finetune LLaMA-7B with RLHF on the StackExchange dataset [Lambert et al., 2023] to output helpful answers to technical questions, as judged by humans. Users ask technical questions on StackExchange and upvote the best answers. We score answers from StackExchange based on the number of upvotes they received from users, $\texttt{score} = \log_2(1 + \texttt{upvotes})$ [Askell et al., 2021] . At most 10 answers are drawn per question, text is cleaned, and HTML is converted to Markdown to make it easier to parse. First, we finetune LLaMA-7B with language modelling on the dataset to get LLaMA-7B-SE. We then further finetune it to get a reward model by learning to predict which of two answers was more upvoted [Stiennon et al., 2020]. For a given question $x$ and two answers $y_{+,-}$ (where $y_+$ is preferred), the loss for our reward model $r_\theta$ is $\log \sigma(r_\theta(x, y_+) - r_\theta(x, y_-))$. Finally, we finetune LLaMA-7B-SE with RL against the reward model by sampling questions from the dataset and learning to optimize the reward for our model's answer. All finetuning is done with a converted 8-bit model [Dettmers et al., 2022] and LoRA [Hu et al., 2021] for efficiency and to make the model training fit on our GPUs. We rely on the HuggingFace trl [von Werra et al., 2023] and peft [Mangrulka et al., 2023] libraries. All technical details are described in Appendix C.

**Experiment**   We again compare to PPO with Ziegler et al. [2019] modifications i.e. KL penalty with a dynamically decaying coefficient. We run for 600 epochs (equivalent to 100k updates) and Elastic Reset every 260 epochs to get two resets / three iterations. Each run takes 20 hours on 4 A100s. Since only the LoRA parameters are being learned, we use Elastic Reset on those and therefore maintain only a small percentage of parameters in our EMA. We use a decay rate $\eta = 0.995$ and a KL penalty coefficient $\beta = 0.02$ for both methods. Calculating perplexity for each epoch is computationally

Table 3: Evaluations of the initial (zero-shot) and finetuned StackLLaMA models after 600 epochs. We measure alignment using an average over three reward model trained with three different seeds and drift with perplexity on the data. HumanEval is a programming benchmark that acts as a practical measure of drift / alignment tax.

| | ↑ Δ Reward | ↓ Perplexity | ↑ HumanEval (pass@1,pass@10) |
|---|---|---|---|
| Zero-shot | 0 | 4.43 | 11.0, 12.7 |
| PPO | $0.81 \pm 0.06$ | 4.62 | 7.8, 10.7 |
| **Elastic Reset** | $0.96 \pm 0.09$ | 4.57 | 11.0, 13.0 |

infeasible so we measure drift during training with the KL from the pretrained model over samples as done previously in other larger-scale RLHF [Bai et al., 2022, Gao et al., 2022].

**Results** We plot reward in Figure 6a [2] and KL from initial model over training in Figure 6b. As noted by Beeching et al. [2023], the task is much noisier at a larger scale and with a real HF reward model. As a sanity check, we find that Elastic Reset tracks PPO until the first reset at 260 epochs where it drops only slightly, but also doesn't lose much performance. Around the second reset at 520 epochs, we see a much sharper drop but also maintaining the same approximate reward. At the end, Elastic Reset provides a non-trivial reduction in drift while aligning just as well as PPO. The pareto curve in Figure 6c shows Elastic Reset is equal or slightly worse at low reward but shows large improvements over PPO at higher reward. Notably, Elastic Reset seems to work out-of-the-box with LoRA. To evaluate drift another way, we get the perplexity of the final models over the StackExchange validation set as in Section 6. For a more robust view of reward, we train two more reward models using different seeds and evaluate the increase in reward between initial and final models. We show mean and standard deviation across the three reward models in Table 3, we find that Elastic Reset achieves a slightly better final reward than PPO while maintaining lower perplexity on the data. To examine a true alignment tax, we run our models on HumanEval Chen et al. [2021], a programming benchmark that provides another view of drift. Answering human-written coding questions is both a useful capability for our model and also falls within a similar domain to StackExchange. The benchmark tests for functional correctness such that pass@1 corresponds to the percentage of problems solved by the model on the first try as shown in Table 3. Training with PPO degrades performance compared to the initial model, demonstrating a large alignment tax. In contrast, Elastic Reset achieves a similar reward but maintains performance, even slightly improving on pass@10, creating an alignment bonus [Askell et al., 2021] instead of tax.

## 8 Limitations

As a method, Elastic Reset is quite cheap computationally because both EMA updates and resets take neglagable time compared to RLHF training and the EMA model can be stored on CPU. But our method is sensitive to the choice of reset rate; we chose heuristically based on when it seemed the model was overfitting. It is also possible to reset the policy and EMA model at different time scales, which could be a source of improvement. Our method also resets all of the trainable parameters, research in similar methods suggests that resetting larger models can benefit from resetting only part of the network [Zhou et al., 2022, Nikishin et al., 2022] or weighted-averaging instead of resets [D'Oro et al., 2023]. We leave both of these directions to future work.

Although we have thoroughly investigated our method on three different tasks, we note that none of them are ideal RLHF benchmarks. As pointed out by Gao et al. [2022], we measure our model's performance using the same reward model we optimize. This can lead to reward model overoptimization and our metric could mask overfitting and lack of generalization to the real world i.e. actual human preferences. An ideal benchmark could include a "gold" reward model as a proxy for human preference [Gao et al., 2022], but no such benchmarks are open-sourced and available.

---

[2]Note that our reward results differ from the original StackLLaMA likely due to a pending issue in their code affecting their reward model. After fixing the issue, the only difference seems to be an increase in base reward, and our reward curves are visually quite similar to theirs. See Appendix C for details

Finally, we note that we follow all previous RLHF work and investigate only on-policy methods [Ziegler et al., 2019, Stiennon et al., 2020, Askell et al., 2021, Bai et al., 2022]. Previous work in resetting for RL has focused on off-policy methods and demonstrated strong performance [Nikishin et al., 2022, D'Oro et al., 2023]. As previously noted, our method can be seen as an adaptation of those to on-policy RL. In RLHF, PPO is by far the most popular method and on-policy is the dominant paradigm since it guarantees better local gradients. But it is possible that off-policy methods could implicitly balance performance and drift by incorporating a replay buffer with older data.

## 9  Conclusion

The problems of drift [Lee et al., 2019], alignment tax [Askell et al., 2021], reward model overoptimization [Gao et al., 2022], and reward hacking [Clark and Amodei, 2016] are inherent to RLHF and reduce its efficacy. We have introduced a simple but powerful new method, Elastic Reset, to tackle this problem and improve performance while maintaining linguistic capabilities. We have shown its ability on three different tasks and across three different scales: from 6 layer Transformers to GPT2 to LLaMA-7B. The problem of drift is currently being addressed with a standard KL penalty despite the computational cost, tradeoff with reward, and recent claims that is may be unnecessary [Bai et al., 2022, Gao et al., 2022]. Elastic Reset is a cheap and effective method to tackle the same problem, achieving a better tradeoff of reward and drift while reducing alignment tax. We hope our method leads to better RLHF and therefore models that are closer aligned with human preferences [Ziegler et al., 2019]. As well, we hope this work invigorates more research into improving the reward / drift tradeoff of RLHF with a focus on computationally efficient methods that scale.

## Acknowledgments and Disclosure of Funding

MN is supported by Fonds de recherche du Québec – Nature et technologies and Sony. MN would like to thank Issam Laradji, ServiceNow Research, Mila, and Compute Canada for providing resources used in the experiments.

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

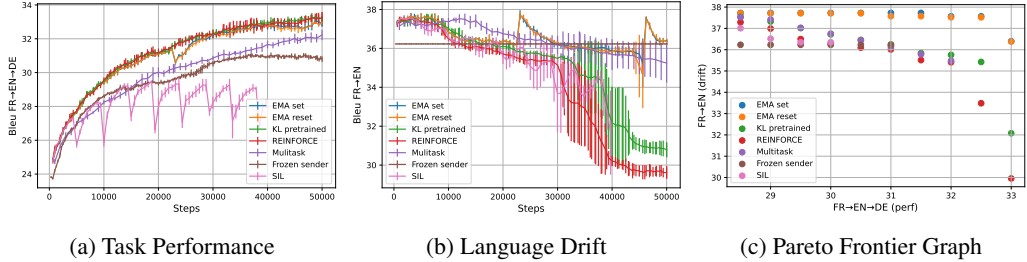

| (a) Task Performance | (b) Language Drift | (c) Pareto Frontier Graph |

Figure 7: All transformer model methods on the Translation Game. We measure (a) Task Performance with FR→EN→DE BLEU and (b) Language Drift with FR→EN BLEU, on the validation set during finetuning. We plot the mean and show error bars for standard deviation over 5 seeds. To compare how methods do on both metrics, we plot (c) the best achieved drift vs task performance across finetuning.

# A  Translation Game

## A.1  Experimental Details

We implement the translation game in the fairseq library [Ott et al., 2019] on top of Pytorch [Paszke et al., 2019]. All experiments are run with 5 seeds where each run uses a single 16G V100 GPU. All plots show the mean and standard deviation over seeds. We score BLEU using detokenized sacreBLEU [Post, 2018]. Plots are made with Matplotlib [Hunter, 2007]

## A.2  Translation Game with Transformers

Note that Lee et al. [2019] (as well as previous work on the Translation Game) use seq2seq [Sutskever et al., 2014] LSTMs [Hochreiter and Schmidhuber, 1997] with attention [Bahdanau et al., 2015]. For consistency with other experiments and relevance to mainstream research we switch to a Transformer architecture. LSTMs results are similar and results are available in Appendix A.3. Also note that we use REINFORCE as our estimator but Gumbel-Softmax [Jang et al., 2017, Maddison et al., 2017] is also feasible [Lu et al., 2020] and preliminary results have been similar.

We follow Lee et al. [2019] for preprocessing both IWLST and Multi30k. We use the `iwslt-de-en` architecture from the fairseq library [Ott et al., 2019] with their default IWSLT DE-EN hyperparameters and training, specifically training with AdamW [Loshchilov and Hutter, 2022] using default hyperparameters. We pretrain our models on IWSLT following Lee et al. [2019] and early-stop on the tst2013 validation set. Our final validation scores are 43.85 BLEU for FR→EN and 29.4 BLEU for EN→DE.

We report all final validation scores in Table 4 and all curves in Figure 7.

Given that we have a learning reward model (EN→DE), iteratively resetting our FR→EN model to its pretrained weights can be an effective strategy [Rita et al., 2022]. We report this as RESET and find it performs quite well but not as well as Elastic Reset. We did not include these results in the main paper because we focused on methods that could also work with fixed reward models.

Table 4: Translation Game final validation scores

|  | ↑ FR→EN→DE | ↑ FR→EN |
|---|---|---|
| PRETRAINED | 23.8 | 36.2 |
| FROZEN SENDER | 30.8±0.2 | **36.3±0.1** |
| REINFORCE | **33.2±0.3** | 29.6±0.3 |
| + SIL | 28.2±0.4 | 27.3±4.4 |
| + MULTITASK (S2P) | 32.2±0.3 | 35.2±1.0 |
| + KL PRETRAINED | **33.2±0.2** | 30.8±0.4 |
| + RESET | 32.5±0.1 | 33.3±0.1 |
| + RESET TO EMA | 32.9±0.1 | **36.3±0.1** |
| + ELASTIC RESET | 33.0±0.1 | **36.3±0.1** |

## A.3 Translation Game with LSTMs

We follow Lee et al. [2019] for preprocessing and pretraining on IWSLT. We compare our pretraining results to both Lee et al. [2019] and Lu et al. [2020] in Table 5

Table 5: BLEU score of IWSLT-pretrained LSTM models on IWSLT 2013 validation set

|  | FR→EN | EN→DE |
|---|---|---|
| LEE ET AL. [2019] | 34.1 | 22.0 |
| LU ET AL. [2020] | 32.2 | 20.2 |
| OURS | 38.5 | 23.2 |

Next we finetune on Multi30k using the same hyperparameters as Lu et al. [2020]. We plot results in Table 6 and compare to published numbers from previous work. As usual, drift is the negative change in FR→EN BLEU from the pretrained model. Task performance is the positive change in FR→EN→DE BLEU from the pretrained models. Combined is the sum of drift and task performance. We show the pretrained, baseline, and best-performing model from previous work. Note that our results are not directly comparable to previous work because we evaluate using detokenized sacreBLEU [Post, 2018] whereas previous work wrote their own BLEU evaluation code and did not detokenize.

Table 6: BLEU scores and ± standard deviation on the Multi30k Translation Game using IWSLT-pretrained LSTM models.

|  | METHOD | FR→EN | FR→EN→DE | DRIFT | PERF | COMBINED |
|---|---|---|---|---|---|---|
| LEE ET AL. [2019] | PRETRAINED | 27.2 | 16.3 | | | |
| | REINFORCE | 12.4 ± 0.7 | 24.5 ± 1.5 | -14.8 | +8.2 | -6.6 |
| | + LM | 23.6 ± 1.1 | 27.7 ± 0.4 | -3.6 | +11.4 | +7.8 |
| | + LM + G | 24.8 ± 0.4 | 28.1 ± 0.7 | -2.4 | +11.8 | +9.4 |
| LU ET AL. [2020] | PRETRAINED | 29.4 | 15.7 | | | |
| | GUMBEL-SOFTMAX | 14.5 ± 0.8 | 27.1 ± 0.1 | -14.9 | +11.4 | -3.5 |
| | SIL | 29.4 ± 0.3 | 28.3 ± 0.2 | 0 | +12.6 | +12.6 |
| **OURS** | PRETRAINED | 32.6 | 18.0 | | | |
| | FROZEN-SENDER | 32.6 ± 0 | 25.1 ± 0.1 | 0 | +7.1 | +7.1 |
| | REINFORCE | 30.0 ± 0.2 | 30.3 ± 0.2 | -2.6 | +12.3 | +9.7 |
| | LM $\lambda = 0.01$ | 31.5 ± 0.1 | 30.2 ± 0.2 | -1.1 | +12.2 | +11.1 |
| | MULTITASK $\lambda = 0.1$ | 32.9 ± 0.2 | 28.5 ± 0.3 | +0.3 | +10.5 | +10.8 |
| | SIL | 27.7 ± 0.7 | 24.3 ± 0.1 | -4.9 | 6.3 | +1.4 |
| | ELASTIC RESET | 32.6 ± 0.1 | 30.0 ± 0.1 | 0 | +12.0 | **+12.0** |

Our results for the baseline are notably better than Lee et al. [2019], Lu et al. [2020]. The only difference between our code and theirs as far as we can tell is (1) we use an exponential moving average baseline for REINFORCE whereas [Lee et al., 2019] use an Actor-Critic method, (2) Lu et al. [2020] uses 0.1 gradient clipping and we do not use gradient clipping (3) in preprocessing Multi30k, we first tokenize [Koehn et al., 2007] then lowercase whereas previous works did the opposite order.

A more reasonable explanation for the improvement in results is how we choose to evaluate. Previous work simply ran all methods for the same number of updates but this doesn't account for, even implicit, hyperparameter optimization. Previous methods show that the baseline's FR→EN→DE BLEU scores plateau and there is significant language drift in FR→EN without real improvements to the task score. We hypothesize that these extra training episodes only serve to increase the drift without measuring what we actually care about: performance gain for drift. Since the number of updates is arbitrary, we believe that early stopping on a reasonable metric is a better evaluation protocol and choose hyperparameters such that methods plateau at the end.

We also note that our results with the SIL method of Lu et al. [2020] are negative. We do not manage to gain any improvement in performance. We collaborated with the authors of Lu et al. [2020] for many months but, in our setup using the fairseq library [Ott et al., 2019] could not reproduce their results. One of their fundamental results is that a student sender can outperform a teacher sender that it is distilling from. We could not reproduce this and believe it is a difference in the pretrained models.

# B    IMDB Mock Sentiment

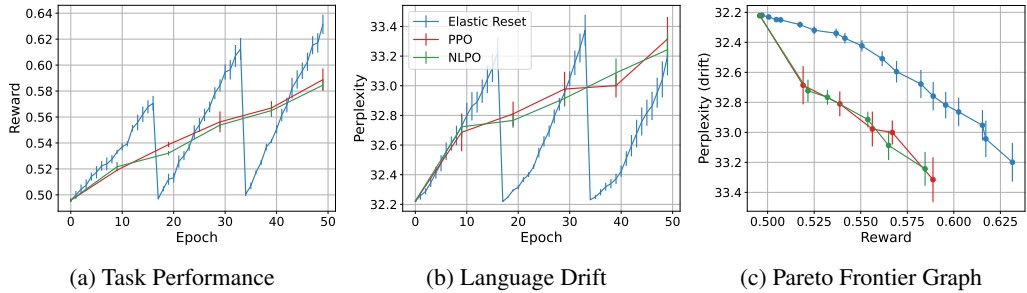

(a) Task Performance        (b) Language Drift        (c) Pareto Frontier Graph

Figure 8: IMDB mock sentiment validation set scores for Elastic Reset, PPO, and NLPO. We measure (a) Language Drift and (b) Task Performance via Semantic Score on the validation set over finetuning. All methods also include a KL penalty. We plot mean and standard error across 5 seeds.

## B.1    Experimental Details

We run experiments and implement our method in the RL4LMs library [Ramamurthy et al., 2022]. All experiments are run with 5 seeds where run uses a single 40G A100 GPU. All plots show the mean and standard error over seeds.

For PPO, we use the default hyperparemters provided by Ramamurthy et al. [2022]. We also compared to NLPO, we found the defaults had a mistake and after communication with Ramamurthy et al. [2022] we changed the learning rate to 1e-6 and target update iterations to 50. We plot results in Figure 8. This still did not manage to reproduce the original NLPO test scores from Ramamurthy et al. [2022] but we found that our validation curves matched their provided curves in the appendix. After communications [Ammanabrolu, 2023], we both agreed that we should use our reproducible test scores for NLPO in lieu of the original test scores.

Our best Elastic Reset hyperparameters are the default PPO parameters (`gpt2_ppo.yml`) from RL4LMs [Ramamurthy et al., 2022] with a few modifications: target kl is set from 0.5 to 1.0 and KL coefficient is set much lower to 0.001. We use an EMA decay of 0.995 and reset every 17 epochs. Configs to reproduce the experiments can be found in the RL4LMs folder under `scripts/training/task_configs/imdb_text_continuation`

# C    StackLlama

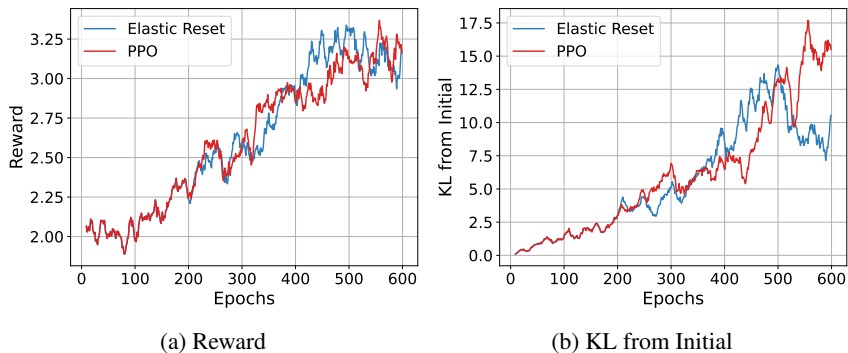

(a) Reward        (b) KL from Initial

Figure 9: Rolling Window Mean (window = 10) version of the StackLLaMA results

## C.1    Experimental Details

We follow the original StackLlama and use the trl library from Huggingface on top of the transformers library [Wolf et al., 2020] to train the model on the StackExchange dataset, loaded with datasets

[Lhoest et al., 2021]. We use the original authors' LoRA adapter weights to create our supervised model LLaMA-7B-SE but train our own reward model as there were issues loading the pretrained reward model. As noted by [Beeching et al., 2023], the reward modelling task is difficult enough that humans struggle with it and our final model achieves 64% accuracy compared to the original 67%. Although we follow the original authors method and use their codebase, we note that our results may be different but valid. There have been many updates and fixes to the `trl` codebase since the authors' original blog post and specifically a possible issue in the code for creating reward models could have affected the original authors' run.

Furthermore, to speed up training, we used a 2x smaller KL coeffient and ran with half the number of GPUs. Specifically, RL training is run on 4x 80G A100 GPUs for 20 hours using Accelerate [Gugger et al., 2022]. We evaluate perplexity of the model on the validation set used in supervised finetuning, 4000 examples from the supervised training dataset. The configs for reproducing our experiments are in the `trl` library folder under `examples/stackllama/scripts/configs`

For visual clarity, we provide a version of our StackLLaMA results with a rolling window mean in Figure 9

## C.2 HumanEval

Our measure of alignment tax, HumanEval [Chen et al., 2021], is a programming benchmark where each question was hand-written by humans (OpenAI engineers) to be unseen in training data. We believe these questions were unseen in LLaMA training data as well. We prompt the model with the question and it writes the corresponding code. The measure of success is functionally correct code i.e. we actually execute the code LLaMA wrote and see if it gets the right output. Since we decode by sampling, we generate $N = 100$ continuations of the prompt and then follow Chen et al. [2021] to estimate how often our model would get the right answer on the first generated continuation (pass @ 1) and within the first 10 generated continuations (pass @ 10). We evaluate HumanEval using CodeCapybara's evaluation harness [To et al., 2023].

# D  Ablations Full Graphs

## D.1  Reset Number

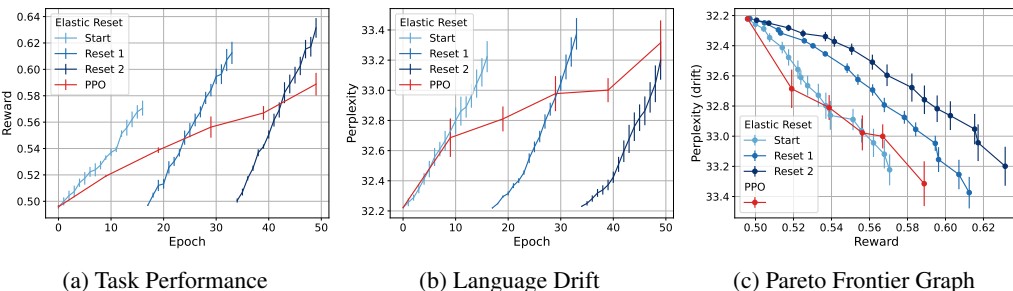

(a) Task Performance    (b) Language Drift    (c) Pareto Frontier Graph

Figure 10: Plotting PPO vs Elastic Reset on IMDB but splitting the results visually between resets. Users with less compute can choose fewer resets. We didn't find improved results after 2 resets for IMDB

## D.2 Reset Type

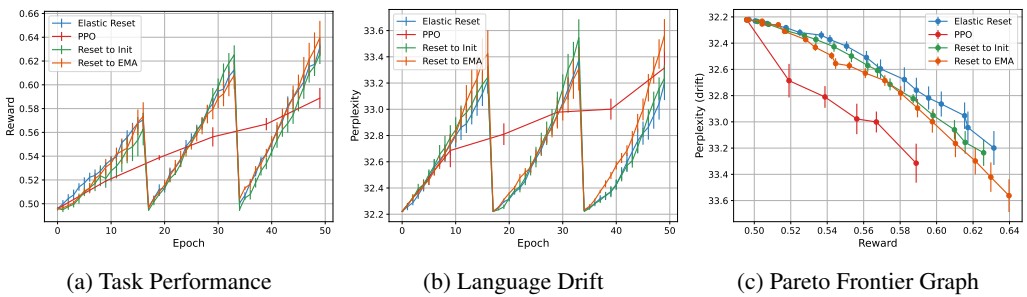

(a) Task Performance

(b) Language Drift

(c) Pareto Frontier Graph

Figure 11: Ablation of Elastic Reset, Reset to EMA, and Reset to Init on IMDB. We plot the mean and show error bars for standard error over 5 seeds.

## D.3 KL Coefficient

As shown in Figure 5, Elastic Reset performance in relatively unaffected by the presence or absence of KL. In contrast, PPO and REINFORCE performance is known to be sensitive [Lu et al., 2020]. This raises the question of whether different coefficients of the KL loss would lead to different pareto frontiers. We find in Figure 12 that both Multitask and KL with Pretrained methods do have slightly different pareto frontiers. Choosing a

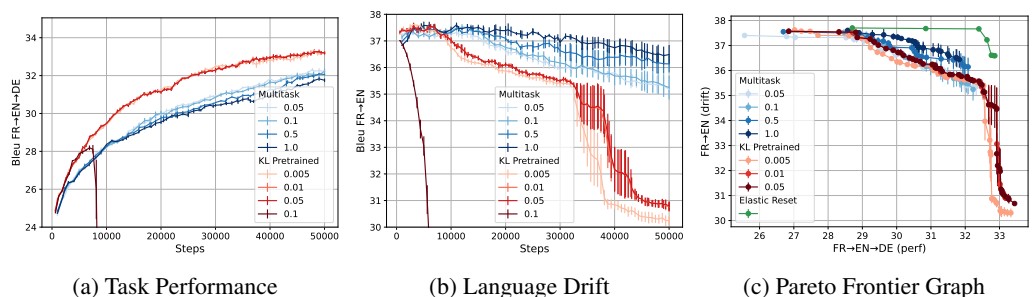

(a) Task Performance

(b) Language Drift

(c) Pareto Frontier Graph

Figure 12: Ablation of KL and Multitask coefficients on Translation Game. We exclude KL 0.1 from the pareto graph since it fails. We plot the mean and show error bars for standard error over 5 seeds.

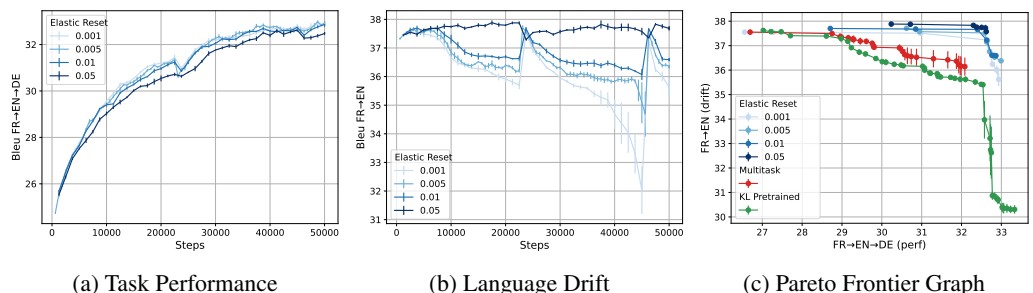

(a) Task Performance

(b) Language Drift

(c) Pareto Frontier Graph

Figure 13: Ablation of KL coefficients for Elastic Reset on Translation Game. We plot the mean and show error bars for standard error over 5 seeds.

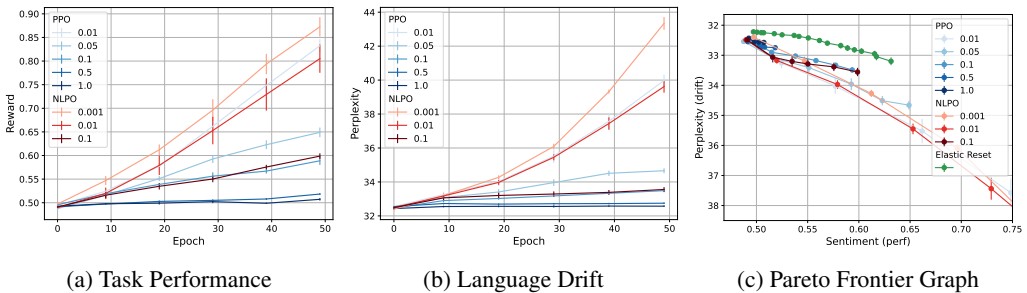

(a) Task Performance     (b) Language Drift     (c) Pareto Frontier Graph

Figure 14: Ablation of PPO and NLPO coefficients on IMDB. We do more ablations of PPO since NLPO is always similar but worse than PPO. We plot the mean and show error bars for standard error over 5 seeds.

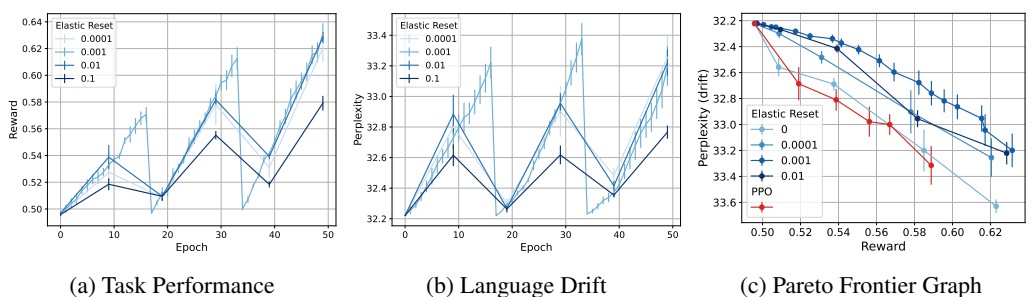

(a) Task Performance     (b) Language Drift     (c) Pareto Frontier Graph

Figure 15: Ablation of KL coefficients for Elastic Reset on IMDB. We plot the mean and show error bars for standard error over 5 seeds.

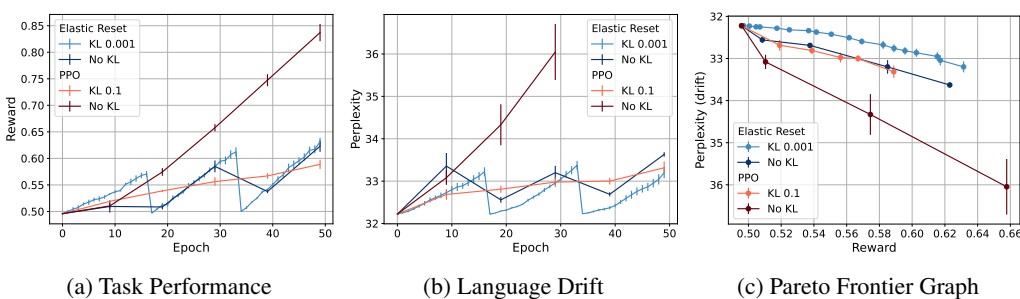

(a) Task Performance     (b) Language Drift     (c) Pareto Frontier Graph

Figure 16: Ablation with/without KL penalty for PPO and Elastic Reset on IMDB. We plot the mean and show error bars for standard error over 5 seeds. For clarity, we cut off PPO without KL to four data points on the pareto graph since it diverges too much

## D.4 Decay Rate

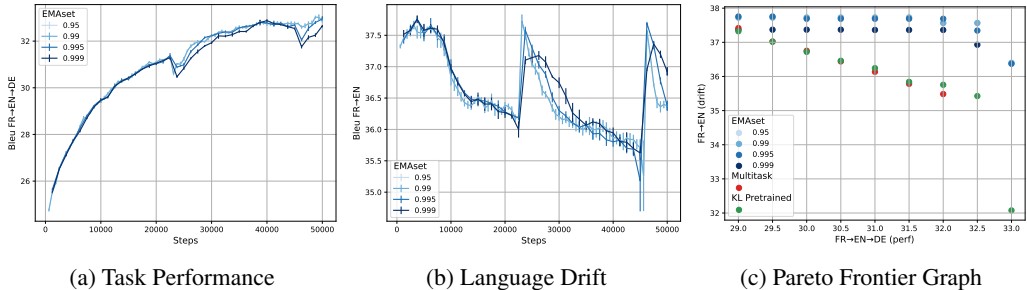

(a) Task Performance          (b) Language Drift          (c) Pareto Frontier Graph

Figure 17: Ablation of decay coefficients for Elastic Reset on Translation Game. Original decay is 0.99. We plot the mean and show error bars for standard error over 5 seeds.

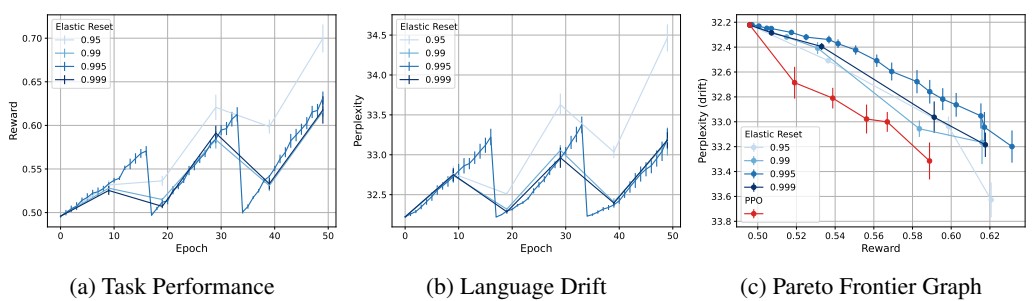

(a) Task Performance          (b) Language Drift          (c) Pareto Frontier Graph

Figure 18: Ablation of decay coefficient for Elastic Reset on IMDB. Original decay is 0.995. We plot the mean and show error bars for standard error over 5 seeds.

### D.4.1 Reset Frequency

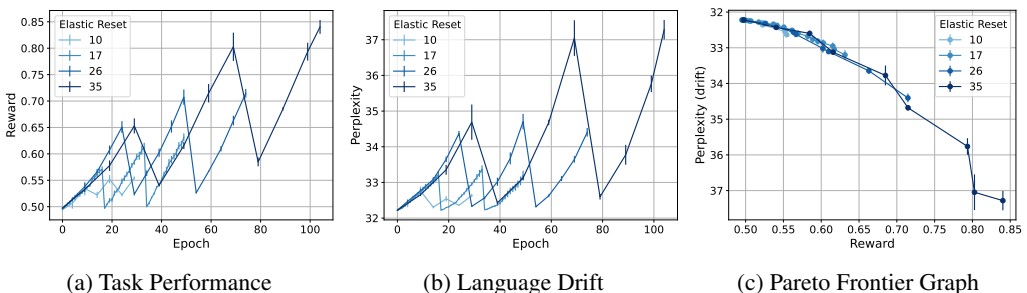

(a) Task Performance          (b) Language Drift          (c) Pareto Frontier Graph

Figure 19: Ablation of reset frequency for Elastic Reset on IMDB. We plot the mean and show error bars for standard error over 5 seeds. Although 17 is optimal, results are relatively robust to choice of this hyperparameter

## E  Explaining Elastic Reset

Here we provide empirical intuition for why Elastic Reset is effective

### E.1  Explaining Resets: Value Function

Drift is a problem of noisy optimization, and Elastic Reset tackles this in two ways.

The first is by re-training with an improved value function. Though we begin with a sequence-level reward model, over training we learn a *token-level* value function. We believe that the value

function greatly improves over time so early training may have exacerbated drift because it uses the early, sub-optimal value function. With each reset of the policy while maintaining the value function, we re-train with less noise and more direct gradients to alignment. Another way, both the reward model and optimal value function point towards high reward, but not in the same way. The reward model points towards high reward but not necessarily in the most direct way. Using the frozen reward model as our value function leads to equivalent reward but higher drift compared to using an optimal value function.

To demonstrate this phenomenon, we compared what happens if we don't train our value function at all but use a frozen reward model as our value function. Since our value function is GPT-2 but our reward model is DistilBERT, we first train a GPT-2 reward model similar to DistilBERT. We then compare regular PPO with a frozen value function i.e. just the sequence-level reward model vs a training value function. In Figure 20 we find that the learning value function is clearly better than the frozen one. They both reach the same reward but a more optimal value function leads to less drift for the same reward.

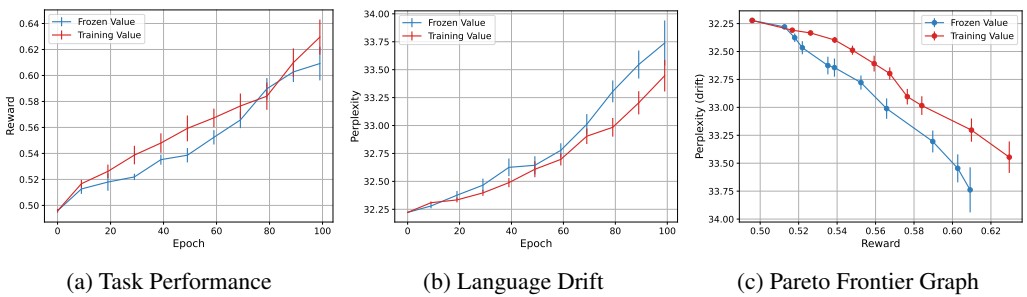

(a) Task Performance      (b) Language Drift      (c) Pareto Frontier Graph

Figure 20: Ablation of learning vs frozen value function for PPO on IMDB.

## E.2   Explaining Elastic Reset: The Benefit of EMA

The second is by smoothing optimization with an EMA. Work in other fields e.g. DINO in SSL, has leveraged EMA for stability and improved generalization. By resetting to an EMA and resetting our EMA, we further smooth the gradients to alignment. To demonstrate, we run PPO as usual but keep track of its EMA, without resetting to it. We then plot the accuracy of the online and EMA networks over training. To make the effect larger, we use a smaller KL $\beta = 0.01$ and train our PPO for 100 epochs and plot results in Figure 21. We find that just keeping the EMA model is an effective way to improve the task / drift tradeoff. But we also see that it is noticeably slower than our method. PPO - Online needs to reach nearly the maximum possible reward for its EMA to improve only slightly on performance.

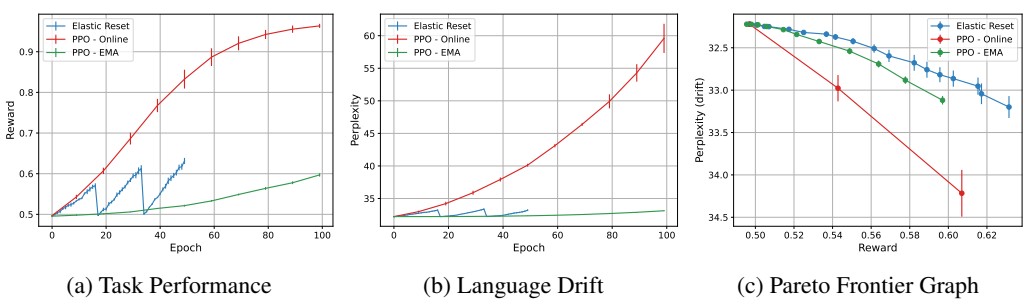

(a) Task Performance      (b) Language Drift      (c) Pareto Frontier Graph

Figure 21: PPO trained on IMDB with a lower KL coefficient for 100 epochs, comparing its online network performance vs EMA model performance. We cut off PPO - online to just three data points in the pareto graph for clarity and visual scale.

