# OpenReview forum: "Language Model Alignment with Elastic Reset"
_NeurIPS.cc/2023/Conference — NeurIPS 2023 poster_

### Official Review · Reviewer_8rTL · 2023-07-04

**Soundness:** 3 good
**Presentation:** 3 good
**Contribution:** 3 good
**Rating:** 6
**Confidence:** 4

**Summary:**

This paper proposes a new approach for fine-tuning language models (LMs) with RL in order to achieve a good trade-off between maximizing reward and minimizing drift from the initial model, which is typically undesirable since it results in losing some capabilities while acquiring others. The approach consists in resetting the model to an exponentially moving average (EMA) of itself, while also resetting the EMA model to the initial one. The method is simple and demonstrates good performance on three different settings with varying degrees of complexity, including RLHF on top of LLaMA-7B for a QA task.

**Strengths:**

1. The proposed approach is simple and effective.

2. The evaluation is quite thorough considering multiple settings and thus demonstrating the generality of the approach.

3. The paper is well written and the problem setting is clearly stated.

4. The problem setting is of high importance to the community. The authors point out a major limitation with existing RLHF methods and propose a method for improving upon it.


**Weaknesses:**

1. The main limitation of this study is the evaluation protocol used to assess the results. The model's performance is evaluated by the same reward model used to train it. However, it is well known that LMs fine-tuned with RL(HF) are prone to overoptimizing their reward (models), overfitting and actually performing badly when evaluated by humans (which is a more robust metric and ultimately what we care about for many applications). While I understand that performing human evaluations can be expensive, it is very difficult to assess the validity of these results otherwise. It could be the case that Elastic Reset is a more powerful optimization approach that overoptimizes the reward better than PPO i.e. can obtain high reward during training but this performance doesn't actually transfer well to unseen prompts when evaluated by humans. At the very least, I suggest using a different reward model for evaluation such as a different base model (of similar size) trained on the same data or the same model trained on a different dataset such as the summarization data from [1] or the HHH dataset from [2]. You could also hold out part of your data and train a separate reward model with a different base on it in order to bring it more in-domain.

2. Can you include experiments with varying reset intervals for the all tasks? It's important to know how sensitive the model is to this parameter and better understand if similar / same values work across different tasks. Do you have any suggestions or insights for selecting this hyperparameter for new tasks?

3. It would also be interesting to see how the results change if only some of the parameters are reset e.g. the last few layers as it typically done when fine-tuning LLMs.

References:
[1]. Stiennon et al. 2020, Learning to summarize from human feedback.

[2]. Bai et al. 2022, Training a Helpful and Harmless Assistant with Reinforcement Learning from Human Feedback.

**Questions:**

1. Why don't you show results with the same KL coefficients for PPO and Elastic Reset in Figure 5?
 Do smaller values of KL not work at all for PPO? What about larger values for Elastic Reset?

2. You mention that Elastic Reset may also work without the KL penalty but you never test this hypothesis. Can you include an experiment with Elastic Reset and KL coefficient of 0? For a fair comparison, it would also be interesting to compare with PPO without the KL penalty, which I assume will perform much worse.

3. Figure 6 is rather noisy, can you include a smoother version in the appendix to better see how the two methods compare with each other?

4. In Figures 5 and 6, the light blue line can barely be seen on printed paper, can you please use a different / stronger color?

**Limitations:**

Yes, the authors clearly state the limitations of their study at he end of the paper.

---

> ### Author Rebuttal · Authors · 2023-08-10
>
> Thank you for the review, we are glad you find the approach simple and effective, the writing clear, and the problem of high importance to the community. We agree that drift is a major limitation for RLHF methods and wish to address it robustly.
>
> **Testing with another Reward Model**
>
> We agree that evaluation is key and find the proposed evals interesting but hope to demonstrate why our protocol is already robust to reward-hacking. Testing with models trained on other reward datasets (e.g. HHH) doesn’t make sense for any of our tasks since they are optimizing very different rewards (StackExchange helpfulness is good coding != HHH Helpfulness which is friendliness / general QA. Summarization is just human-preferred summaries). Testing with a new reward model trained on held-out examples is possible but has not been done in previous literature and would deviate our results from existing baselines (by using a different training reward model without held-out examples).
>
> We think our protocol is robust to reward hacking as
> 1. The pareto graph measure is robust
>
>  A method that does better on the pareto graph must achieve higher reward while staying closer to the original model (lower KL) which, by definition, is reducing drift. By staying closer to the original model, it is likely to better transfer, maintain more of the original model’s ability, and less likely to be over-optimized (reward-hacking)
>
> 2. External tests for IMDB and StackLLaMA show robustness
>
> We evaluate on a set of separate test prompts for the IMDB task (Table 2) and demonstrate that our method outperforms the baselines there, transferring well to unseen prompts. On StackLLaMA, we use a separate coding benchmark HumanEval that should be correlated with good StackExchange answers, but is completely separate from our reward model. We find better coding performance with the Elastic Reset-trained model, clearly showing that it has learned better coding, not reward hacking.
>
> **Varying Reset Interval**
>
> We provide new results on IMDB varying the reset interval from 10, 17, 26, and 35 epochs, keeping everything else constant. We plot new results in Figure 3 in the additional pdf attached in the global response at the top. As shown in the pareto curve, Elastic Reset is robust to choice of reset interval, we simply chose 17 to fit two resets into 50 IMDB epochs.
>
> Sadly, we could not run this for StackLLaMA as it takes too long for this rebuttal period.
>
> **Elastic Reset vs PPO with same KL**
>
> As shown in Figure 5c, PPO diverges quickly with KL 0.01, and does even worse with smaller KL coefficients
>
> **Experiments without KL**
>
> We ran both PPO and Elastic Reset without KL penalty in IMDB and plot results in Figure 2 in the additional pdf. PPO diverges quickly but Elastic Reset without KL actually performs comparable to PPO’s best result with KL.
>
> **Cleaner Figure 6**
>
> We have added smoothing with a rolling mean of window size 10 but could not include it in the additional pdf due to lack of space. We will add this to the appendix for the final paper.
>
> **Partial Resetting**
>
> We agree this idea is interesting and include it as possible future work in Section 8.
>
> **Light Blue is Too Light**
>
> We have made the lightest blue stronger/darker for the final paper, please see the graphs in the additional pdf.

---

> > ### Comment · Reviewer_8rTL · 2023-08-11
> > **Post-Rebuttal Review**
> >
> > I thank the authors for their detailed answers to my questions and appreciate that they ran additional experiments as suggested, which seem to support the claims in the paper.
> >
> > Regarding the reward model used for evaluation, I agree with the authors that the experiments support some degree of generalization, particularly the ones on StackLLaMA. However, I was not convinced by this statement "Testing with a new reward model trained on held-out examples is possible but has not been done in previous literature and would deviate our results from existing baselines (by using a different training reward model without held-out examples)." First of all, more and more papers are using multiple models for evaluation ([1], [2], [3] to name a few) so I expect this to soon become standard practice, and in any case it provides a more robust evaluation. What I'd propose is to use two different reward models, the one that you already have which would allow easy comparison with existing literature, and another one that would strengthen the robustness of your evaluations.
> >
> > I also agree with reviewer tYGJ's comment that evaluating the approach on more challenging tasks beyond the IMDB one would further strengthen the paper and make the results more convincing and relevant to practical applications.
> >
> > In conclusion, I am willing to increase my score to 6 conditioned that the authors commit to including evaluations using another reward model to ensure the results are robust to this choice.
> >
> > References:
> > [1]. AlpacaFarm: A Simulation Framework for Methods that Learn from Human Feedback. Dubois et al. 2023.
> > [2]. LIMA: Less Is More for Alignment. Zhou et al. 2023.
> > [3]. Secrets of RLHF in Large Language Models Part I: PPO. Zheng et al. 2023

---

> > > ### Author Response · Authors · 2023-08-14
> > > **Results with more Reward Models**
> > >
> > > We followed the reviewer's suggestion and train two more reward models exactly as our first but with different random seeds. We evaluate LLaMA 7B after supervised finetuning zero-shot, and after RLHF with PPO and Elastic Reset as in Table 3. We measure the change in reward from the initial model reward and show the average and standard error across the seeds. We also add change in perplexity for reference.
> > >
> > > | | $\Delta$ Reward $\uparrow$ | $\Delta$ Perplexity $\downarrow$ |
> > > | --- | --- | --|
> > > | Zero-shot | 0.00  | 0
> > > | PPO | 0.81 $\pm$ 0.06  | 0.19
> > > | Elastic Reset | 0.96 $\pm$ 0.09 | 0.14
> > >
> > > We agree this is a more robust eval and thank the reviewer for the suggestion. We would like to note the novelty of this approach as all three papers the reviewer referenced were released after the NeurIPS deadline and none of them evaluate over multiple trained reward models. LIMA and AlpacaFarm evaluate with OpenAI APIs not trained reward models, though AlpacaFarm does use multiple prompts to approximate multiple annotators. Zheng et al (2023) do train two reward models but they are used separately for separate languages (English and Chinese) as far as we can tell.

---

> > > > ### Comment · Reviewer_8rTL · 2023-08-16
> > > > **Response to Additional Experiments**
> > > >
> > > > I appreciate that you ran additional experiments with multiple reward models by changing the seed. It's encouraging to see that the results are consistent. As promised, I will increase my score.
> > > >
> > > > However, note that I still believe using different model sizes / architectures or training them on different datasets, rather than merely training them with different seeds, would make the evaluation even more robust.

---

### Official Review · Reviewer_tYGJ · 2023-07-06

**Soundness:** 3 good
**Presentation:** 3 good
**Contribution:** 3 good
**Rating:** 5
**Confidence:** 4

**Summary:**

This paper aims to address the problem of language drift issue during RLHF (reinforcement learning with human feedback), which is also known as alignment tax and reward hacking. The problem is that during RLHF process, the model can "overfit" to the given suboptimal rewards while forgetting some important skills such as linguistic capabilities. The authors proposed a method named Elastic Reset that periodically reset the online model with an exponentially moving average (EMA) of its previous checkpoints. The authors use a few tasks to showcase the effectiveness of the proposed Elastic Reset method on top of GPT-2 and LLaMA-7B, and shows that it is better than vanilla PPO with a reduced KL penalty.

**Strengths:**

The method is rather simple and easy to implement. It just needs to save the parameters of previous checkpoints during training, so that every $n$ steps, one can compute an EMA and merge it to the current model.

The empirical results suggest that the proposed method is effective on the selected tasks and datasets, outperforming the simple PPO and NLPO baseline methods.

**Weaknesses:**

- The method itself is not particularly novel. Using Reset mechanism and EMA of model parameters to mitigate overfitting is rather common, although it might be a novel application in RLHF.

- The "drift" problem is not very clearly defined and formulated. The motivation is not well justified. Many descriptions are references while there is no concrete experiments and case studies to show the problem of drifting clearly.

- I believe the key issue that this paper wants to address is essentially the same to many continual learning problems -- learning new knowledge while not forgetting the acquired skills. Therefore, many CL methods such as experience replay, regularization, EWC (Elastic weight consolidation), should also be applicable. But none of them is mentioned in the paper. The authors focused too much on the RLHF literature, using newly invented terms, however, ignored that the key challenge can be formulated with existing problem setup and can be addressed by existing techniques.

- The selected tasks and datasets are quite narrowed. The experiments on GPT-2 and even smaller models are also not that convincing in that RLHF is rarely used on such LMs. I suggest authors can replace those experiments with more commonly used datasets to support the claims.


**Questions:**

- Line 219, why do you "greatly reduce the KL coefficient $\beta$"? From the description, it seems that you want to amplify the drift issues, but it seems choosing different $\beta$ can significantly influence both PPO and Elastic Reset. How do you decide choosing which $\beta$ to compare your method and other baselines? Do you have to decide this before you see the real test data?

**Limitations:**

The authors use Section 8 to describe the limitations.

---

> ### Author Rebuttal · Authors · 2023-08-10
>
> We are glad the reviewer finds our method simple and effective. We hope the following clarifies the method and importance of our work.
>
> **Novelty**
>
> We are not aware of any method that resets to an EMA to counter overfitting, and we can’t find any method that also resets the EMA model as we do with Elastic Reset. We would be happy to cite others and explain how our method compares if the reviewer can give references.
>
> **Definition and Importance of “Drift”**
>
> We define drift as “performance degradations of a language model as a result of RL finetuning and improvement on a reward objective”, see [Lazaridou et al (2020)](https://arxiv.org/abs/2005.07064) for a full taxonomy of drifts. The phenomenon also known as “alignment tax” is generally agreed to be a major downside of RLHF finetuning [(Askell et al, 2021)](https://arxiv.org/abs/2112.00861) and many previous works that we cite have demonstrated the issue in toy tasks [(Lee et al, 2019)](https://arxiv.org/abs/1909.04499) to real-world RLHF chatbots [(Bai et al, 2022)](https://arxiv.org/abs/2204.05862).
>
> **Connection to Continual Learning**
>
> We agree there are many links to CL. We focus on RLHF-specific terms, known methods, and benchmarks to make our work most useful to the many RLHF projects currently in progress. We are happy to add a summary of CL connections to our related work as well as other references the reviewer may have:
>
> The pretrain-then-RL-finetune setup with the goal of maintaining pretrained knowledge can be seen as a two-step, RL-specific instance of continual learning and therefore language drift has links to catastrophic forgetting (McCloskey & Cohen, 1989). There is a clear similarity between mitigation methods: rehearsal (Robins, 1995) or experience replay (Rolnick et al, 2019) is equivalent to multitasking with the pretraining objective (Lowe* et al., 2021) and weight-update regularization (Kirkpatrick et al., 2017) has similarities to KL regularization (Jaques et al, 2019).
>
> **Are Tasks Appropriate?**
>
> The IMDB task is the most popular RLHF benchmark, used as a test-bed for many papers including the original work [(Ziegler et al, 2019)](https://arxiv.org/abs/1909.08593). We use the standard version from the GRUE benchmark [(Ramamurthy et al, 2022)](https://arxiv.org/abs/2210.01241) because it includes a strong, tuned PPO baseline and allows us to demonstrate trends and do quantitative evaluations that are reasonable given our compute.
>
> **Choosing Hyperparameters**
>
> Following the GRUE benchmark, we chose all our hyperparameters on the validation set of the IMDB and test on the test set (Table 2). StackLLaMA was too expensive to tune so we simply applied what we knew from IMDB for our hyperparameters.
>
> **Elastic Reset uses smaller KL coefficient**
>
> Because PPO drifts so much, it requires a very high KL coefficient and does not work well otherwise. Elastic Reset, accounts for drift by resetting so if you use a high KL coefficient, resets will not be useful within IMDB’s 50 epochs as there will not be much drift to counter. We choose an optimal, smaller KL so that there is more drift for Elastic Reset to counter.
>
> We compare baselines against our own method and choose the best KL coefficient for each within a fixed compute budget i.e. 50 IMDB epochs in the GRUE benchmark. In Figures 5c,d we also show that Elastic Reset works across a wide range of smaller KL coefficients that PPO doesn’t and all of them outperform PPO’s best setting. We find that Elastic Reset even works without KL and is competitive with PPO+KL's best setting (see Figure 2 in the additional pdf included in the global response at the top)

---

> > ### Comment · Reviewer_tYGJ · 2023-08-17
> >
> > Nice explanation and I have raised my score accordingly. Thanks!

---

> ### Author Response · Authors · 2023-08-14
> **Please Review Rebuttal**
>
> Can the reviewer please read and respond to the rebuttal? We have run new experiments that we believe address all the noted issues with the paper and may merit an increase in score. If there are any outstanding issues, we would like the chance to respond before the discussion period is over. Thank you.

---

### Official Review · Reviewer_Xquz · 2023-07-07

**Soundness:** 3 good
**Presentation:** 4 excellent
**Contribution:** 4 excellent
**Rating:** 6
**Confidence:** 4

**Summary:**

This paper proposes Elastic Reset, a simple technique for countering language drift and reward model overfitting when optimizing a language model policy against some communicative reward via reinforcement learning, as is done in RLHF. The idea of elastic reset is to periodically reset the trained model to an exponentialy-weighted moving average (EMA) between the initial pretrained model and the online model trained since the last reset.

The authors demonstrate on a variety of RLHF tasks from relatively simple (pivot translation) to relatively substnatial (StackExchange QA) that Elastic Reset seems to be a simple way to mitigate language drift which outperforms other approaches, including the commonly-used KL penalty in RLHF. They propose a nice way of interpreting the tradeoff by using a "pareto frontier graph", which plots methods' downstream task reward and measure of language drift on x/y axes, similarly to ROC curves, and demonstrate that for some given compute budget, Elastic Reset dominates existing baselines like REINFORCE and PPO with KL penalties.

Overall there's a lot I like about this paper, but there are some issues I have with the experimental setup that prevent me from unconditionally recommending the paper for acceptance. If my concerns are clarified or resolved, I am willing to update my score and look forward to the author response.

**Strengths:**

- A simple technique that appears to give gains across a variety of RLHF benchmarks at varying scales (from translation to stackexchange Q/A). I am impressed by the breadth of experiments in this paper. It seems this could be one of those simple tricks that practitioners find useful and widely deploy in future RLHF pipelines (but time will tell whether the results are robust enough).
- Good comparison to other sensible baselines, with some sensible ablations on the IMDB mock sentiment task, but some areas for improvement here (see Weaknesses)
- I like the Pareto figures, reminiscent of ROC curves, which demonstrate the tradeoff between task performance and language drift and how to identify an optimal method under a given practitioner's constraints. I agree with authors that this is the right way to think about language drift, though I have some points of confusion (see firsrt Weakness)

**Weaknesses:**

## Unclear how robust elastic reset under different compute constraints

- The IMDB and StackLLaMA experiments demonstrate something subtly worrying in my view: they show that there are portions during training where elastic reset *doesn't* help over comparable baselines. For example, in Figure 4 a/b, training prior to the first two resets demonstrate a "spiking" behavior where the model overfits to the reward and has higher language drift than the other two methods. It is only after the 2nd reset, untnil the end of training, that the reset causes task reward to rise higher than existing methods *but*.
	- This feels "lucky" to me, and it is hard to measure how robust elastic reset is under different compute constraints. For example, lets say we only have enough compute to train the model up to (but not after) the 2nd reset, i.e. epoch ~33 in Figure 4. It seems like in this case we would *not* prefer to use the elastic reset model, since it seems to still be in the regime of overfitting, and it is only after the 2nd reset that things start to look better.
- A similar concern exists for StackLLaMA where there is not a significant difference between Elastic reset and PPO, until around ~500-600 epochs, when there suddenly is (and perhaps there's even a regime in 400-500 where Elastic Reset is overfitting).
- To address this concern, it seems like the pareto graphs, and in general the performance of elastic reset, need some notion of compute budget to more accurately assess when it is appropriate to use elastic reset. For example, what does the pareto frontier graph look like if we only consider training up to about epoch 33 in Figure 4? (i.e. before the 2nd reset)? Is elastic reset still the preferred choice? Would changing the number of resets change anything? (Perhaps I'm misinterpreting the pareto graphs here).
- A more detailed analysis of how many resets are needed and how robust elastic reset is to the timing of rests would partially alleviate these concerns (see next point)

## Could use more carefully controlled baselines

- The baselines could be clearer. If my understanding is correct, the *only* way in which elastic reset diverges from traditional RLHF is by periodically resetting the model to be the EMA of the past n model steps. As authors say, this is a strength of the method, but the comparisons tend not to directly compare to a model with/without elastic reset. For example, considering Elastic reset with 3 resets during training, the sensible baseline is to compare to the exact same method (e.g. REINFORCE), with the same hyperparameters, just with 0 resets during training. But the baselines in the paper seem to always have a slight confounding factor, e.g. for the pivot translation task L152, a KL penalty for Elastic Reset is added on top of the REINFORCE baseline that does not seem to be present in vanilla REINFORCE; same for L219 on top of PPO (the beta parameter is reduced, and the decision to let the PPO model "drift more" is not well justified); only the StackLLAMA experiment in Section 7 appears to stay consistent (L280), but the improvement of elastic reset here is not as clear.
- More generally it would be better if, instead of a "1 vs many" comparison where a single implementation of elastic reset is compared to a single implementation of PPO and REINFORCE baselines, a "paired" comparison strategy was adopted, where for each method and specification of hyperparameters for each baseline, authors measure the effect of elastic reset on top (as authors say, it seems simple to just add this onto any arbitrary alignment technique, even not necessarily RLHF). Does Elastic Reset improve consistently over other RLHF methods, keeping hyperparameters consistent? Of course, it doesn't have to improve all the time, but knowing when it helps and when it doesn't (because maybe the baseline method by itself keeps language drift under control) would be nice.
	- Again, Figure 5c is lacking some context. The Elastic Reset figure is built on PPO with beta = 0.01. I appreciate the KL ablation but does Elastic Reset dominate PPO across all KL penalty values kept constant? Otherwise what is the reason for setting the KL penalty lower for elastic reset?
- Again, to address this concern, I would love an investigation of how many resets affect performance, and any guidelines for choosing a set number of resets, in the same style as the nice ablations in Figure 5. Authors state that it is difficult to find a heuristic for how often to reset (L307), but graphs showing the effect of performance for different reset timescales and/or compute would be enormously informative, and help address the first main weakness I outlined above.

## Mathematical connection/intuition as to why elastic reset differs from distillation and/or KL

- I get the feeling there are some mathematical connections to both KL penalty and alternating RL/distillation algorithms that are not fully explored in the paper. I haven't thought too deeply, but, for alternating RL/pretraining methods (S2P; Lowe et al., 2021) and iterated distillation, one can think of the distillation/SL phase as precisely doing a sort of model reset to the initial pretrained model via gradient descent to minimize KL between the online model and the initial model. For RLHF with KL penalties, we can also view this as a sort of bayesian inference, computing a model average of the prior pretrained model and the posterior (online) model trained via RL ([Korbak et al., 2022](https://arxiv.org/abs/2205.11275)). The number of steps taken is a hyperparameter that, if chosen carefully, results in a model after distillation that is likely some average of the online model and the initial pretrained model, as in elastic reset. If elastic reset is a simpler or more efficient way of doing the same thing as such a distillation phase, this is a great thing, but it's not quite clear to me now. Could authors clarify any differences between distillation and elastic reset? In particular, why might we expect elastic reset to be **more** performant than than alternating RL/distillation or a KL penalty, given that they seem to have the same motivation (which might even be mathematically formalized)?

## Minor
- L99 "Elastic Reset takes inspiration from both of these"—it's not quite clear what "both" refers to.
- Some qualitative examples of model outputs in the appendix could potentially be nice to have in the paper, perhaps even identifying what model outputs look like at different points on the pareto frontier curve for different training methods.

**Questions:**

Main questions in Weaknesses section. Some more minor questions:
- L100 "EMA on CPU"—are authors sure this is more efficient? If EMA is on CPU, don't you need a full model copy from GPU to CPU for each step? This seems expensive, and suggests that Elastic Reset would also benefit from keeping the EMA on the GPU. (You could theoretically keep the pretrained model used for KL penalties on the CPU and copy over, but this would significantly slow things down)
- Authors could add some intuition (or experiments) that discuss how important is it that the reset resets to an **exponentially-weighted** moving average? Why not just a literal model average between the new model and the pretrained model?

**Limitations:**

yes

---

> ### Author Rebuttal · Authors · 2023-08-10
>
> Thank you for the detailed review! We are glad you find our method simple but effective and tested against sensible baselines across a breadth of experiments. We also believe pareto curves are the right way to think about language drift and hope that our work increases their adoption as a standard in RLHF evaluation.
>
> **Compute Constraints**
>
> Our method does require 2 resets to outperform baselines but notably uses the exact same compute as the baseline to achieve these results. This is in line with all other works using resetting that frequently require many resets to outperform baselines.
>
> We also agree that compute is an interesting axis of comparison so we’ve created a new graph where we plot each run between resets as its own pareto curve so practitioners can choose at which reset to stop (see Figure 1 in additional pdf attached in the global response). Thank you for the suggestion. A user with less compute can also make resets more frequent to get lower reward but less drift (see Figure 3 in additional pdf).
>
> **Baselines**
>
> We always compare to the exact model without Elastic Reset (i.e. 0 resets)
> - KL Penalty (Reinforce + KL) in Translation Game
> - PPO (includes KL Penalty) in IMDB
> - PPO (includes KL Penalty) in StackLLaMA
>
> **Fair baselines use different KL coefficients**
>
> Because PPO drifts so much, it requires a very high KL coefficient and does not work well otherwise. Elastic Reset accounts for drift by resetting so if you use a high KL coefficient, resets will not be useful within IMDB’s 50 epochs as there will not be much drift to counter. But Elastic Reset can still work with that large KL coefficient: simply train for more than 50 epochs to get more drift and reset less frequently (e.g. every 50 epoch instead of 17).
>
> For a fair comparison, we limit both runs to the same compute. So following all previous work, we choose the best KL coefficient for each baseline and our own method within a fixed compute budget i.e. 50 epochs for IMDB following the [GRUE benchmark](https://rl4lms.apps.allenai.org/grue). In Figures 5c,d we also show that Elastic Reset works across a wide range of smaller KL coefficients that PPO doesn’t and all of them outperform PPO’s best setting.
>
> **Choosing Number of Resets**
>
> On IMDB, we tested resetting 1,2,3,5,and 10 times but found no improvement over 2. We hope the new reset-pareto graph described above helps choose the number of resets and allows future work to demonstrate the tradeoff.
>
> **Elastic Reset Intuition**
>
> Distillation is one possible way to view the EMA and so Elastic Reset can be seen as a compute-efficient alternative to Iterated Learning (which alternates learning and distillation). We can see that an EMA model drifts less than its online model (see Appendix D.5) which could imply it is an effective distillation for RLHF.
>
> Another reason why Elastic Reset works is it doesn’t reset the value function. So after each reset, the model starts with a better initialization (EMA) but also better value function that can more directly point towards high reward (see Appendix D.4). This also helps explain the improvements after each reset in Figure 1 in the additional pdf.
>
> **EMA on GPU vs CPU**
>
> It is indeed easier to keep EMA on GPU and we do this for our experiments. If memory is tight, it is possible to do EMA updates every $n$ steps instead of every step and therefore the GPU to CPU copy is less frequent. On IMDB, we found updating the EMA every 100 steps achieves similar results to updates every step.
>
> **Weighted average instead of EMA**
>
> We use an EMA since it is a simple and efficient update with each gradient update and our own experiments suggest it mitigates drift (Appendix D.5). Simple weighted averages are an interesting idea and will likely work but we leave it for future work as we don’t expect any improvements or advantages over EMA.

---

> > ### Comment · Reviewer_Xquz · 2023-08-15
> > **Thanks**
> >
> > Thanks to authors for the response. Upon looking at Figure 4a/b closely it seems like there are indeed still regimes where under 1 or 2 resets, doing elastic reset allows one to get higher reward with less drift. Figure 1(c) in the supplement is helpful for illustrating this point and would be useful to include in the supplementary material.
> >
> > The other responses addressing questions regarding elastic reset intuition, KL penalties, etc are also helpful.
> >
> > I'm raising my score to a 6.

---

> ### Author Response · Authors · 2023-08-14
> **Please Review Rebuttal**
>
> Can the reviewer please read and respond to the rebuttal? We have run new experiments that we believe addresses all the noted issues with the paper and may merit an increase in score. If there are any outstanding issues, we would like the chance to respond before the discussion period is over. Thank you.

---

### Official Review · Reviewer_1Gya · 2023-07-10

**Soundness:** 3 good
**Presentation:** 3 good
**Contribution:** 3 good
**Rating:** 7
**Confidence:** 4

**Summary:**

Finetuning language models with reinforcement learning (RL) using human feedback, i.e. RLHF, has emerged as a promising paradigm for aligning large language models (LLM) to human preferences. Though RLHF has shown promising results when training models such as ChatGPT, RL has some inheritance drawbacks. Merely optimizing a reward model trained on human preferences can degrade performance, known as reward hacking, alignment tax, or language drift in the literature. This paper argues that the standard way for addressing this is insufficient and proposes a new idea. In particular, the authors propose, Elastic Reset, a technique to reset the model weights to address reward hacking.

**Strengths:**

Addressing reward hacking is an age-old problem, and many solutions have been proposed. The strength of this paper is that the solution proposed has several key benefits:
- Easy to implement: The authors implemented their idea on top of several existing RLHF frameworks and showcased the benefit of Elastic Reset to existing frameworks RL algorithms.
- Written Presentation: The author's presentation of the proposed algorithm, explanations, and ablations studies were justified and thoroughly explained.
- Experiments: The authors presented several experiments across three very different domains and showcased the benefit of their proposed method.
- Clear Definition of Language Drift: Often language drift is not clearly defined, but found very clear examples to showcase language drift issues.

**Weaknesses:**

Although the proposed algorithm is straightforward to implement and has shown good results across three challenging datasets, the approach has several weaknesses.
- Resetting on-policy vs. off-policy: Resetting weights are typically done with off-policy algorithms because the policy has a problem exploring when increasing the replay ratio. Whereas for on-policy to deal with exploration, we typically add temperature parameter, entropy loss coefficient, or some other exploration bonus. Instead, in the on-policy case, the authors use reset to keep the policy close to the original policy so it does not experience language drift, which seems like the opposite use-case of the original intent.

- Second reset: The first reset in which EMA is reset to the initial model means that you are still searching around an epsilon ball around the initial model. I am unsure why this would be much better than KL divergence, which explicitly does this.

**Questions:**

- How did you decide on resetting of the first stage? Given stage one, how did you decide on resetting the second stage?

- Why did you include the supervised learning (SL)+PPO results in your experiments? Training RL from scratch is known to be a difficult task. Most RLHF success stories warm start the RL model using SL. The results in the paper seem to use the RL with PPO results from the GRUE benchmark. But SL+RL results are always stronger.

[Algorithm] - Sentiment Score - Perplexity

[PPO +SL] - 0.626   - 35.045

[NLPO + SL.] - 0.611    - 33.82

Is the proposed approach is not compatible with RL+SL?

**Limitations:**

Yes

---

> ### Author Rebuttal · Authors · 2023-08-10
>
> Thank you for your review! We’re glad you found our problem compelling and clearly defined, our method easy to implement, and our experiments thorough in demonstrating the benefit of our method.
>
> **On-policy Elastic Reset vs prior work off-policy Reset**
>
> A major difference between our work and prior work in resets is that we reset to an EMA whereas prior works (like Nikishin, Schwarzer et al, 2022) reset to a new random initialization. Resetting to random can indeed help with exploration, but we don’t think that is the benefit of our method. Instead we believe that
> 1. After a reset, the value function is maintained and provides a better direction, see Appendix D.4 for an in-depth summary. Supporting this, we find that each run after a reset is better than the last as shown in Figure 1 in the additional pdf at the top.
> 2. The EMA model drifts less than the online model because it smoothes out the gradient steps, see Appendix D.5. Just using an EMA is too slow, though, and it is difficult to achieve high reward with an EMA alone. Iteratively resetting to an EMA strikes a good balance.
>
> **Elastic Reset vs KL Divergence**
>
> KL simply pulls all gradients towards the pretrained model. Not all change from the pretrained model is bad, as argued by [Gao et al (2022)](https://arxiv.org/abs/2210.10760) there are good gradient directions to move in, even outside the epsilon ball. Elastic Reset stays close to the model but the EMA also seems to provide an inductive bias against drifting in non-linguistic directions (see Appendix D.4)
>
> **Hyperparameter Choices**
>
> We chose all our hyperparameters on the validation set of the IMDB and then applied them to StackLLaMA. We tested resetting 1,2,3,5,and 10 times but found no improvement over 2. We tested resetting the EMA model 2,4x less and more frequently but found no improvement over the same reset schedule.
>
> **SL + PPO**
>
> We do include SL + PPO for the appropriate tasks (Translation Game and StackLLaMA). We don’t do SL+PPO for IMDB following the GRUE benchmark that found stronger results without SL [(Ramamurthy et al, 2022)](https://arxiv.org/abs/2210.01241).

---

> ### Author Response · Authors · 2023-08-14
> **Please Review Rebuttal**
>
> Can the reviewer please read and respond to the rebuttal? We have run new experiments that we believe addresses all the noted issues with the paper and may merit an increase in score. If there are any outstanding issues, we would like the chance to respond before the discussion period is over. Thank you.

---

> > ### Comment · Reviewer_1Gya · 2023-08-16
> >
> > I want to thank the authors for performing additional experiments.
> >
> > **On-policy Elastic Reset vs prior work off-policy Reset**
> > Thank you for the explanation.
> >
> > **Elastic Reset vs KL Divergence**
> > Thank you for the explanation.
> >
> > **Hyperparameter Choices**
> > Thank you for the explanation.
> >
> > **SL + PPO**
> > In the GRUE benchmark (table 3), we see that PPO+Supervised has a sentiment score of 0.626 and a perplexity score of 35.049, whereas PPO has a sentiment score of 0.602 and a perplexity score of 33.816. Furthermore, table 2 shows that RL+Supervised is always better than Supervised, which is not the case for RL without supervised warm starting.
> >
> > [1] IS REINFORCEMENT LEARNING (NOT) FOR NATURAL LANGUAGE PROCESSING: BENCHMARKS, BASELINES, AND BUILDING BLOCKS FOR NATURAL LANGUAGE POLICY OPTIMIZATION by Ramamurthy et al. 2023

---

> > > ### Author Response · Authors · 2023-08-17
> > > **IMDB Clarifications**
> > >
> > > For the IMDB benchmark, we do use the strongest baseline "PPO" and it is warm-started. Elastic Reset outperforms both PPO and Supervised+PPO and we'd like to clarify the differences between the latter methods.
> > >
> > > The initial model for the PPO baseline is GPT-2 after supervised finetuning on IMDB. This is exactly what we're doing on our other tasks as well and guarantees the lowest perplexity for our initial model under the data distribution. What Ramamurthy et al (2023) call "Supervised + PPO" further finetunes that initial model on just the positive examples (and therefore already drifts from the distribution of *all* movie reviews).
> > >
> > > "Supervised + PPO" achieves higher reward than PPO but also has much more drift i.e. higher perplexity. In Ramamurthy et al's main paper it is unclear which model is better but in the Appendix Table 5 for an ablation, the same target KL shows PPO outperform Supervised + PPO. With the default hyperparams, we run an extra experiment to achieve a higher reward and perplexity with PPO by training it 2x longer (100 epochs). We find it achieves the same perplexity as Supervised + PPO in 50 epochs but higher reward. Elastic Reset trained similarly 2x longer outperforms both methods.
> > >
> > >
> > > | |Sentiment $\uparrow$ | Perplexity $\downarrow$ |
> > > |--|--|--|
> > > PPO (appendix KL inf) | 0.838 $\pm$ 0.061 |  41.897 $\pm$ 1.806 |
> > > Supervised + PPO (appendix KL inf) | 0.796 $\pm$ 0.004 |  42.916 $\pm$ 1.716 |
> > > |||
> > > PPO (main paper) | 0.602 | 33.816 |
> > > Supervised + PPO (main paper) | 0.626 | 35.049 |
> > > PPO trained 2x longer (ours) | 0.730 $\pm$ 0.002 | 35.093 $\pm$ 0.2 |
> > > Elastic Reset trained 2x longer (ours) | **0.736 $\pm$  0.01** | **34.722 $\pm$ 0.6** |
> > >
> > >
> > >
> > > We hope this clarifies our results, please let us know if you have any questions.

---

### Author Rebuttal · Authors · 2023-08-10

We've run some extra experiments in response to reviewer comments, please find three figures attached:
1. Plotting each reset of Elastic Reset separately
2. PPO and Elastic Reset with best KL vs without KL (coefficient 0)
3. Elastic Reset over a range of reset frequencies

---

### Author Response · Authors · 2023-08-21

We'd like to thank the reviewers for their timely responses and fruitful discussions. We appreciate the time and effort in the process and believe it has made our work clearer. We will be adding all additional experiments and clarifications from these discussions to the final version of our paper.

---

### Decision · Program_Chairs · 2023-09-21

**Decision:**

Accept (poster)

**Comment:**

The paper proposes a method called elastic reset which adds periodic resets (to an exponentially moving average) to RL-based language generation training with the aim of improving the trade-off between reward optimization and language drift. Experiments on pivot-based translation and IMDB review generation show an improved trade-off between language drift and reward (task performance). There is also a small improvement on RLHF training of LLaMA on StackExchange.

The paper is well-written and of interest since it aims to address an important current problem. While method is simple it leads to improvements across multiple tasks. The reviews raised a number of questions related to the robustness of the experimental evaluation: The rebutals mostly addressed those and provided some additional results, while the use of open-source LLMs enables reproducibility. More experiments with large-scale LMs and/or more RLHF tasks would further strengthen the results. The connection between the proposed method and related concepts in continual learning and distillation can also be strengthened. However on balance the paper's contribution is strong enough to be accepted.